# Coevolution-based prediction of key allosteric residues for protein function regulation

Juan Xie[1], Weilin Zhang[2], Xiaolei Zhu[3], Minghua Deng[1,4,5], Luhua Lai[1,2,6]*

[1]Center for Quantitative Biology, Academy for Advanced Interdisciplinary Studies, Peking University, Beijing, China; [2]BNLMS, Peking-Tsinghua Center for Life Sciences at the College of Chemistry and Molecular Engineering, Peking University, Beijing, China; [3]School of Sciences, Anhui Agricultural University, Hefei, China; [4]School of Mathematical Sciences, Peking University, Beijing, China; [5]Center for Statistical Science, Peking University, Beijing, China; [6]Research Unit of Drug Design Method, Chinese Academy of Medical Sciences (2021RU014), Beijing, China

**Abstract** Allostery is fundamental to many biological processes. Due to the distant regulation nature, how allosteric mutations, modifications, and effector binding impact protein function is difficult to forecast. In protein engineering, remote mutations cannot be rationally designed without large-scale experimental screening. Allosteric drugs have raised much attention due to their high specificity and possibility of overcoming existing drug-resistant mutations. However, optimization of allosteric compounds remains challenging. Here, we developed a novel computational method KeyAlloSite to predict allosteric site and to identify key allosteric residues (allo-residues) based on the evolutionary coupling model. We found that protein allosteric sites are strongly coupled to orthosteric site compared to non-functional sites. We further inferred key allo-residues by pairwise comparing the difference of evolutionary coupling scores of each residue in the allosteric pocket with the functional site. Our predicted key allo-residues are in accordance with previous experimental studies for typical allosteric proteins like BCR-ABL1, Tar, and PDZ3, as well as key cancer mutations. We also showed that KeyAlloSite can be used to predict key allosteric residues distant from the catalytic site that are important for enzyme catalysis. Our study demonstrates that weak coevolutionary couplings contain important information of protein allosteric regulation function. KeyAlloSite can be applied in studying the evolution of protein allosteric regulation, designing and optimizing allosteric drugs, and performing functional protein design and enzyme engineering.

*For correspondence:
lhlai@pku.edu.cn

Competing interest: The authors declare that no competing interests exist.

## Editor's evaluation

The manuscript reports on a useful tool to study protein allosteric regulation function. The work is based on inadequate experimental validation of the predicted residues implicated in mediating allosteric signaling. The study highlights the significance of the weak pairwise term for the prediction of the allosteric function.

## Introduction

Allostery commonly refers to one type of distant regulation, that is, a perturbation at one site of a macromolecule can affect the function of another site (*Dokholyan, 2016*), which plays important roles in many biological processes, such as enzyme catalysis (*Tsai et al., 2009*) and signal transduction (*Hilser et al., 2012*). Compared to traditional orthosteric drugs, allosteric drugs have unique

advantages, including higher specificity, fewer side effects, etc. (*Changeux and Christopoulos, 2016*; *Thal et al., 2018*; *Wenthur et al., 2014*). However, optimization of allosteric molecules faces great challenges as allosteric molecules usually have flat structure-activity relationships (flat SARs or shallow SARs, referring to the phenomenon that no robust SARs could be obtained as small modifications may destroy the activity), and higher binding affinity does not always correspond to better activity (*Christopoulos, 2002*; *Jimenez et al., 2012*; *Lewis et al., 2008*; *Lindsley, 2014*; *Nussinov and Tsai, 2012*). In an allosteric pocket, the contribution of each residue to the allosteric effect is different. Bi et al. found that the interactions between allosteric molecules and the target protein can be divided into two types, interactions that only contribute to binding and interactions that contribute to both binding and signaling. Based on these understandings, they rationally designed Tar variants and engineered *Escherichia coli* to sense new ligands by maintaining the interactions responsible for chemotaxis allosteric signaling while changing the interactions responsible only for ligand binding (*Bi et al., 2013*). Nussinov et al. proposed that atoms in allosteric effectors could be divided into anchor and driver atoms. The anchors docked into the allosteric pockets, which allowed the drivers to perform a 'pull' or 'push' action (*Nussinov and Tsai, 2014*). Both drivers and anchors showed specific interactions with their host proteins, with the former mainly responsible for the allosteric efficacy and the latter for binding affinity (*Nussinov et al., 2014*). Therefore, it is important to identify residues in the host proteins that form interactions with allosteric molecules and produce allosteric signaling, which we refer to as key allo-residues, so that allosteric molecules can be optimized and designed based on these key allo-residues. Unfortunately, identifying key allo-residues remains challenging. Currently available computational methods mainly focused on the prediction of allosteric sites (*Ma et al., 2016*; *Qi et al., 2012*; *Wagner et al., 2016*; *Xie et al., 2022*), allosteric pathways (*Botello-Smith and Luo, 2019*; *Lake et al., 2020*), and key residues in allosteric pathways (*Wang et al., 2020*). Kalescky et al. developed the rigid residue scan method to identify key residues for protein allostery, in which multiple molecular dynamics (MD) simulations need to be performed for unbound and bound proteins. As only one residue was regarded as a single rigid body in each simulation, many simulations were necessary, which are computationally expensive and time-consuming (*Kalescky et al., 2015*). Therefore, methods for systematically and rapidly identifying key allo-residues in protein allosteric pockets need to be developed.

During evolution, unrelated residues may evolve independently, while functionally coupled residues coevolve. Coevolution means that when a residue changes, the residues that are structurally or functionally coupled with it will also change accordingly to maintain the overall spatial structure and biological function (*de Juan et al., 2013*). In principle, since homologous sequences record the long-term evolution of a protein family, the coupling pattern between residues can be estimated from multiple sequence alignment (MSA; *Reynolds et al., 2011*).

Various methods have been developed to analyze residue-residue coupling during evolution, which greatly expedite the recent progress of protein structure prediction (*Ekeberg et al., 2013*; *Marks et al., 2011*; *Morcos et al., 2011*). Direct coupling analysis (DCA) is one of such approaches that can remove the indirect correlation between residues and reflect the direct coevolution between residues (*Cocco et al., 2013*). DCA mainly uses methods in statistical physics to infer the pairwise coupling $J_{ij}$ between positions, which can explain the observed correlation between residues in an MSA. In structure predictions, only the top couplings in $J_{ij}$ were used (*Morcos et al., 2011*; *Weigt et al., 2009*). Recent studies showed that the weak, non-contact couplings in $J_{ij}$ are significantly important for the prediction of protein function, although they play as noise in predicting structural contacts. Salinas et al. proposed that the information of allosteric energy interactions is included in the statistics of MSAs and therefore is part of the entire evolutionary constraints (*Salinas and Ranganathan, 2018*). Russ et al. found that the top coupling items in $J_{ij}$ alone cannot effectively reproduce the alignment statistics in the AroQ family or the functional effects of mutations. This implies that protein functions may depend on many weak, non-contact items in $J_{ij}$. Although there is no simple physical explanation for these weak items at present, they seem to represent the collective global evolution of residues, and further research is needed to reveal the significance of these items (*Russ et al., 2020*). Similar findings have been reported in the DCA-based prediction of protein-protein interaction (PPI), where the quality of prediction depends on many weak couplings (*Bitbol, 2018*).

In the present study, we analyzed the evolutionary couplings (ECs) between residues in orthosteric and allosteric sites. We found that weak couplings in $J_{ij}$ contain allosteric information and developed

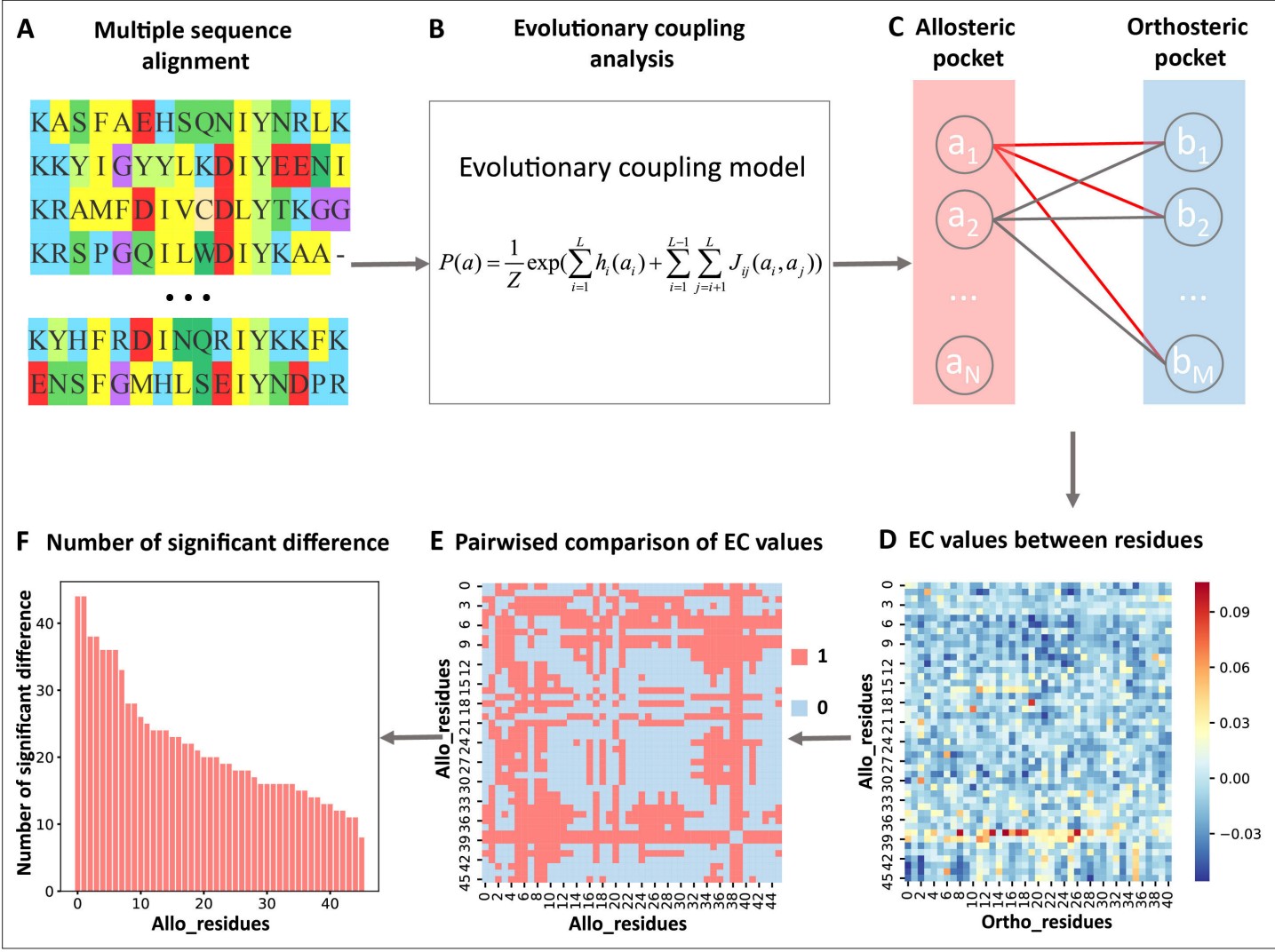

**Figure 1.** Steps to identify key allo-residues. (**A**) Multiple sequence alignment. (**B**) Evolutionary coupling (EC) analysis. (**C–D**) Calculation of the EC values between residues in allosteric and orthosteric pockets. (**E**) Pairwise compared the difference of EC values corresponding to residues in allosteric pocket. (**F**) The number of significant differences corresponding to each residue in allosteric pocket.

the first systematic and efficient computational method KeyAlloSite to predict key allo-residues based on the EC model (ECM; see Materials and methods for details). For each protein in the allosteric protein data set, we first performed MSA (*Figure 1A*) and calculated the ECs between residues using the ECM (*Figure 1B*). We then studied the coevolution between orthosteric and allosteric sites and found that their EC strength (ECS) is stronger than that between orthosteric and other non-functional pockets. We further performed pairwise comparison of the differences in EC scores of residues in the allosteric pocket with orthosteric pocket (*Figure 1C–E*; see Materials and methods for details) and got the number of significant differences corresponding to each residue in the allosteric pocket (*Figure 1F*). After the numbers of significant differences were normalized into Z-scores, residues corresponding to Z-scores larger than a threshold were predicted as key allo-residues. We have applied KeyAlloSite to identify key allo-residues in several allosteric proteins, including BCR-ABL1, Tar, and PDZ3, and compared the prediction results with previously reported experimental data. KeyAlloSite was also used to predict cancer mutations as well as key distant residues for enzymatic catalysis. Our study provides essential information for understanding how allosteric regulations are evolved, for designing and optimizing allosteric drugs, and for designing highly efficient enzymes and other functional proteins.

# Results

## The evolutionary coupling between orthosteric and allosteric sites is stronger

We selected 23 allosteric proteins from the 'Core Set' of ASBench (*Huang et al., 2015*) as our data set, including 25 known allosteric sites (*Supplementary file 1*; see Materials and methods for details). The sequence lengths of the proteins in the data set range from 166 to 788 amino acid residues (*Figure 2A*), and the number of homologous sequences and effective homologous sequences corresponding to each protein were shown in *Figure 2B*. For each protein in the data set, we used CAVITY (*Xu et al., 2018*; *Yuan et al., 2013*) to identify all the potential ligand binding pockets on the protein surface and designated the *m*th pocket as cavity_m. Previous studies showed that motions of orthosteric and allosteric sites are highly correlated (*Ma et al., 2016*; *Xie et al., 2022*; *Zhang et al., 2019*), and the regulation between orthosteric and allosteric site is bidirectional (*An et al., 2019*). It would be interesting to see whether orthosteric and allosteric sites are coupled in evolution. We then explored the ECS ($ECS_{\mathrm{cavity}\_m}$) between the orthosteric and allosteric pocket, as well as all the other pockets. The ECS between the orthosteric pocket and the *m*th pocket is defined as the sum of the coupling strength between the residues in the two pockets (see Materials and methods for details). Among the 25 allosteric pockets in the data set, 23 have Z-scores greater than 0.5 (*Figure 2C*, *Supplementary file 2*), which means that the recall of KeyAlloSite on predicting allosteric sites is 0.92 (*Supplementary file 8*). The probabilities that the known allosteric pockets were ranked in the top 1, top 2, and top 3 of Z-scores were 56.0, 76.0, and 96.0%, respectively (*Figure 2D*, *Supplementary file 2*), indicating that orthosteric and allosteric pockets are more evolutionarily coupled to each other than the orthosteric and other pockets, which can be used to predict potential allosteric pockets. We further analyzed the two proteins with Z-scores less than 0.5, AR1 and CYP3A4. For AR1, as there were only 108 effective homologous sequences, we speculated that the number of homologous sequences may not be enough for evolutionary analysis. The protein sequence–based phylogenetic tree of AR1 homologous proteins showed that AR1 located near the tail of the tree (*Figure 2—figure supplement 1*), implying that this allosteric function did not exist in the early evolutionary period, and it may have appeared in the later stage of evolution. Due to the relatively large number of sequences in the early evolutionary period and relatively few sequences in the later stage of evolution, the allosteric signal was weak. As a cytochrome P450 protein, CYP3A4 can bind and catalyze the transformation of a variety of substrates (*Williams et al., 2004*). Sequence alignments showed that several positions in its orthosteric pocket are less conserved, which may lead to the difficulty in allosteric site prediction based on evolutionary coupling analysis.

We used human Aurora A (AurA) kinase that is not included in the data set as a test case to further verify whether the ECS can be used to predict allosteric sites. AurA (PDB ID: 1OL5) is a Ser-Thr protein kinase that is essential for the cell cycle progression. Its abnormal levels can lead to inappropriate centrosome maturation, spindle formation, and enhanced cancer growth (*Toji et al., 2004*). AurA is known to be regulated by two distinct allosteric mechanisms, one is specific PPI, which binds TPX2 to its hydrophobic pocket, and the other is phosphorylation of the activation loop at T288 (pT288; *Hadzipasic et al., 2020*). We used CAVITY to find all of the potential ligand binding pockets on the surface of AurA, and a total of 12 pockets were found; cavity_3 is the known allosteric PPI pocket, and cavity_2 is the orthosteric pocket. For consistency, we chose residues within 6 Å around the ATP molecule as the orthosteric pocket. Then we calculated the ECS between the orthosteric pocket and each of the remaining 11 pockets. Cavity_3 ranked the second among the 11 pockets with a Z-score of 1.48 (*Supplementary file 3*), indicating that the ECS between the orthosteric and allosteric pockets is indeed stronger. Since phosphorylation mainly occurs on Serine/Threonine/Tyrosine (Ser/Thr/Tyr) residues, we then calculated the ECS between the orthosteric pocket and each of the 26 exposed Ser/Thr/Tyr residues and normalized to Z-scores. Among the 26 Ser/Thr/Tyr residues, 10 residues have Z-scores larger than 0.5, and T288 and T287 ranked the fifth and the fourth with a Z-score of 0.83 and 1.05, respectively (*Supplementary file 3*). This indicates that KeyAlloSite can also be used to predict post-translational modification (PTM) sites, and the predicted Ser/Thr/Tyr residues with Z-scores greater than 0.5 in addition to T288 and T287 are worth further investigation.

To exclude the possible influence of the pocket size, we further checked the dependence of $ECS_{\mathrm{cavity}\_m}$ on the number of residues used in the calculation. For allosteric and other pockets in each

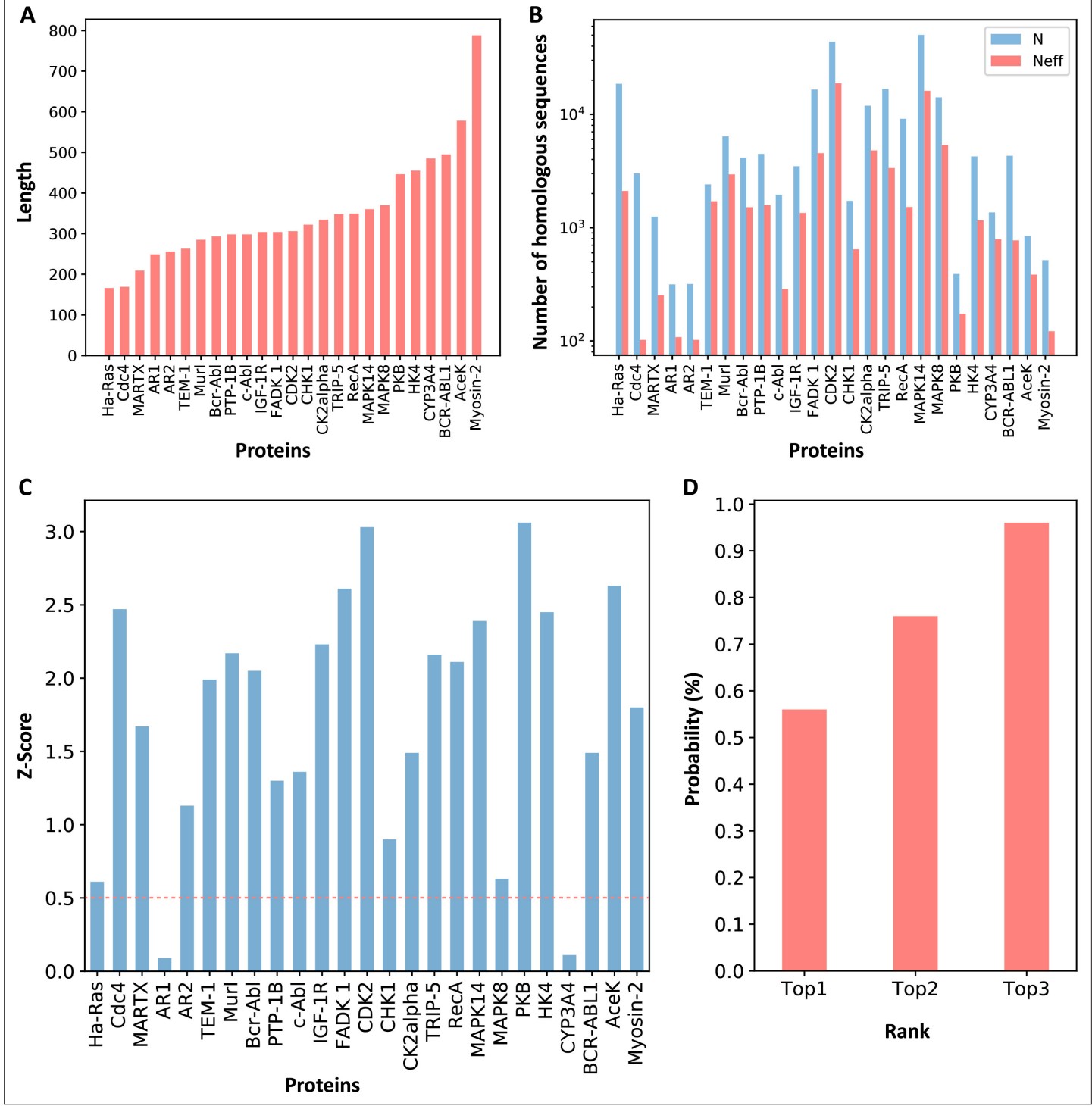

**Figure 2.** Z-scores of allosteric pockets and probabilities of ranking an allosteric pocket in the top 3. (**A**) The sequence lengths of all proteins in our data set. (**B**) The number of homologous sequences. Neff represents the number of effective homologous sequences obtained under 80% reweighting. (**C**) Z-scores of allosteric pockets on proteins in the data set. Among the 25 allosteric pockets, the Z-scores of 23 allosteric pockets were greater than 0.5. (**D**) The probabilities that the known allosteric pockets were ranked in the top 1, top 2, and top 3.

The online version of this article includes the following source data and figure supplement(s) for figure 2:

**Source data 1.** Raw data for *Figure 2*.

**Figure supplement 1.** Phylogenetic tree of the androgen receptor.

**Figure supplement 2.** Comparison of evolutionary coupling strength between pockets when all residue pairs and partial residue pairs were used.

*Figure 2 continued on next page*

*Figure 2 continued*

**Figure supplement 2—source data 1.** Raw data for *Figure 2—figure supplement 2*.

**Figure supplement 3.** Difference between the evolutionary coupling between orthosteric and allosteric sites and the evolutionary coupling between two random patches.

**Figure supplement 3—source data 1.** Raw data for *Figure 2—figure supplement 3*.

protein, we summed the ECS of the top 200, top 300, and top 400 residue-residue pairs with the highest $FN(i,j)$ corresponding to each pocket as the ECS between each pocket (except for orthosteric pocket) and orthosteric pocket. When different numbers of residue-residue pairs were used, the Z-scores corresponding to the ECS between allosteric and orthosteric pockets in most proteins were still greater than 0.5, which is weakly different from that when all residue-residue pairs were used (*Figure 2—figure supplement 2*). This indicates that the number of residues in the *m*th pocket does not play important role on the ECS between pockets. In other words, $ECS_{\text{cavity\_}m}$ revealed the intrinsic EC between allosteric and orthosteric pockets. These results indicate that the ECS between orthosteric and allosteric pockets is stronger than that between orthosteric and other pockets, which can be used to predict potential allosteric pockets.

We further checked whether the coevolutionary signal between allosteric and orthosteric pockets is significantly different from that between two random patches in proteins with the same number of residue pairs. For each protein in the data set, two residues that are not part of the orthosteric and allosteric sites were randomly selected from the surface residues. Among them, one was taken as the first center, and the residues around it with the same number as the residues in orthosteric pocket were selected as patch1; and the other residue was taken as the second center, and the residues around it with the same number as the residues in allosteric pocket were selected as patch2. Then we calculated the ECS between patch1 and patch2. The process was repeated four times, and then the mean and standard deviation of the ECS were calculated. We then compared the ECS between patch1 and patch2 with that between orthosteric and allosteric sites. The results showed that the ECS between orthosteric and allosteric sites was significantly higher than that between two random patches (*Figure 2—figure supplement 3*). In other words, there is intrinsic EC between orthosteric and allosteric sites, which is different from the EC between any two random patches.

## Coevolution analysis revealed key allo-residues in allosteric pockets

For each protein, we calculated the EC values between the residues in the orthosteric and allosteric pockets by ECM and compared the corresponding pairwise EC values of the residues in the allosteric pocket (*Figure 1C–E*, *Figure 3—figure supplement 2*). Since orthosteric and allosteric pockets are two different pockets, the residues in the two pockets generally do not contact. Therefore, most of the EC values between residues in the two pockets are relatively small, that is, they correspond to the weak terms in $J_{ij}$ (*Figure 1D*). We then calculated the number of significant differences of each residue in the allosteric pocket and normalized to Z-scores. Finally, residues were predicted as key allo-residues if their corresponding Z-scores were greater than 0.8 (*Supplementary file 4*). We chose this threshold to ensure that the most known key allo-residues can be correctly predicted by KeyAlloSite, and at the same time, the number of predicted key allo-residues should be as small as possible. We calculated the number of known key allo-residues that could be predicted by KeyAlloSite (*Supplementary file 5*) and the ratios of the predicted key allo-residues in all residues of allosteric pockets in all proteins of the data set (*Figure 3—figure supplement 1*) for thresholds of 0.5, 0.6, 0.7, 0.8, 0.9, and 1.0. Therefore, we finally chose the threshold as 0.8. For the allosteric pockets, the average number of pocket residues is 43, and the average number of identified key allo-residues is 8, accounting for 18.6% of all the residues in the allosteric site (*Figure 3*).

Since the number of homologous sequences is important in coevolution analysis, we selected seven proteins with a relatively large number of homologous sequences and randomly sampled different numbers of homologous sequences. Since the number of homologous sequences required might be related to the sequence length of the protein itself, the number of homologous sequences was divided by the length of the protein to obtain a ratio. Within this ratio, different ratios of homologous sequences were randomly sampled according to different gradients, and each gradient was repeated three times. Then we calculated how many key allo-residues determined by homologous sequences

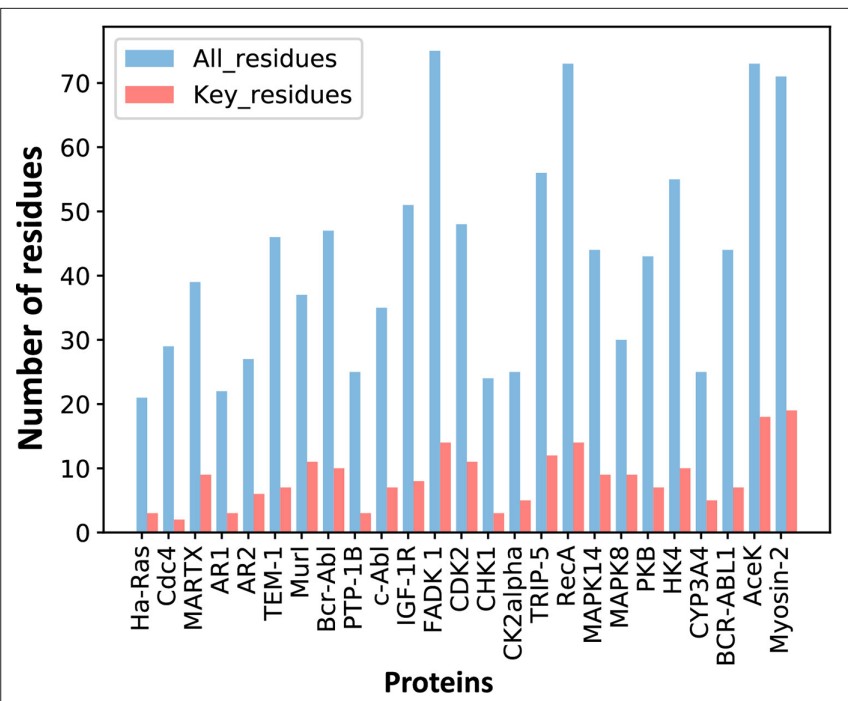

**Figure 3.** The number of predicted key allo-residues. Number of residues refers to the number of residues from allosteric pockets, including the number of all residues in allosteric pockets and predicted key allo-residues.

The online version of this article includes the following source data and figure supplement(s) for figure 3:

**Source data 1.** Raw data for *Figure 3*.

**Figure supplement 1.** Distribution of the ratios of the number of key allo-residues predicted by KeyAlloSite in the number of all residues in allosteric pockets when using different cutoffs in all proteins.

**Figure supplement 1—source data 1.** Raw data for *Figure 3—figure supplement 1*.

**Figure supplement 2.** Examples of distributions of the statistics corresponding to significant scores obtained from the t-test.

**Figure supplement 3.** Random sampling of homologous sequences.

**Figure supplement 3—source data 1.** Raw data for *Figure 3—figure supplement 3*.

---

with different gradients were the same as those determined by all homologous sequences. Taking the key allo-residues determined by all homologous sequences as references, we calculated the proportion of the same residues (*Figure 3—figure supplement 3*). It can be seen that generally speaking, 7 L ($\pm 4L$) number of effective homologous sequences is enough to give good and stable results. If the protein sequence was relatively short, the number of homologous sequences required could be less; otherwise, more homologous sequences were needed.

## The predicted key allo-residues were supported by experimental results

We searched for literatures to see whether the key allo-residues we predicted were experimentally tested before. The first example is tyrosine-protein kinase ABL1 (BCR-ABL1), which is a fusion protein whose constitutive activity can cause chronic myeloid leukemia (CML). Tyrosine kinase inhibitors targeting the ABL1 ATP-binding site, such as imatinib (Gleevec) and nilotinib (Tasigna), significantly improved the overall survival of CML patients (*Kalmanti et al., 2015*; *Miura, 2015*). However, patients may develop drug resistance due to mutations in the ATP-site. The novel fourth generation ABL1 drug, asciminib (ABL001) was developed, which is an allosteric inhibitor that binds to the myristoyl pocket of BCR-ABL1 (*Figure 4A*). Asciminib was developed from fragment-based drug discovery approach. In the early stage of hit identification, compounds that bind BCR-ABL1 without inhibition activity were found. Among them, hit **4** binds BCR-ABL1 with a $K_d$ of 6 μM. After changing the Cl

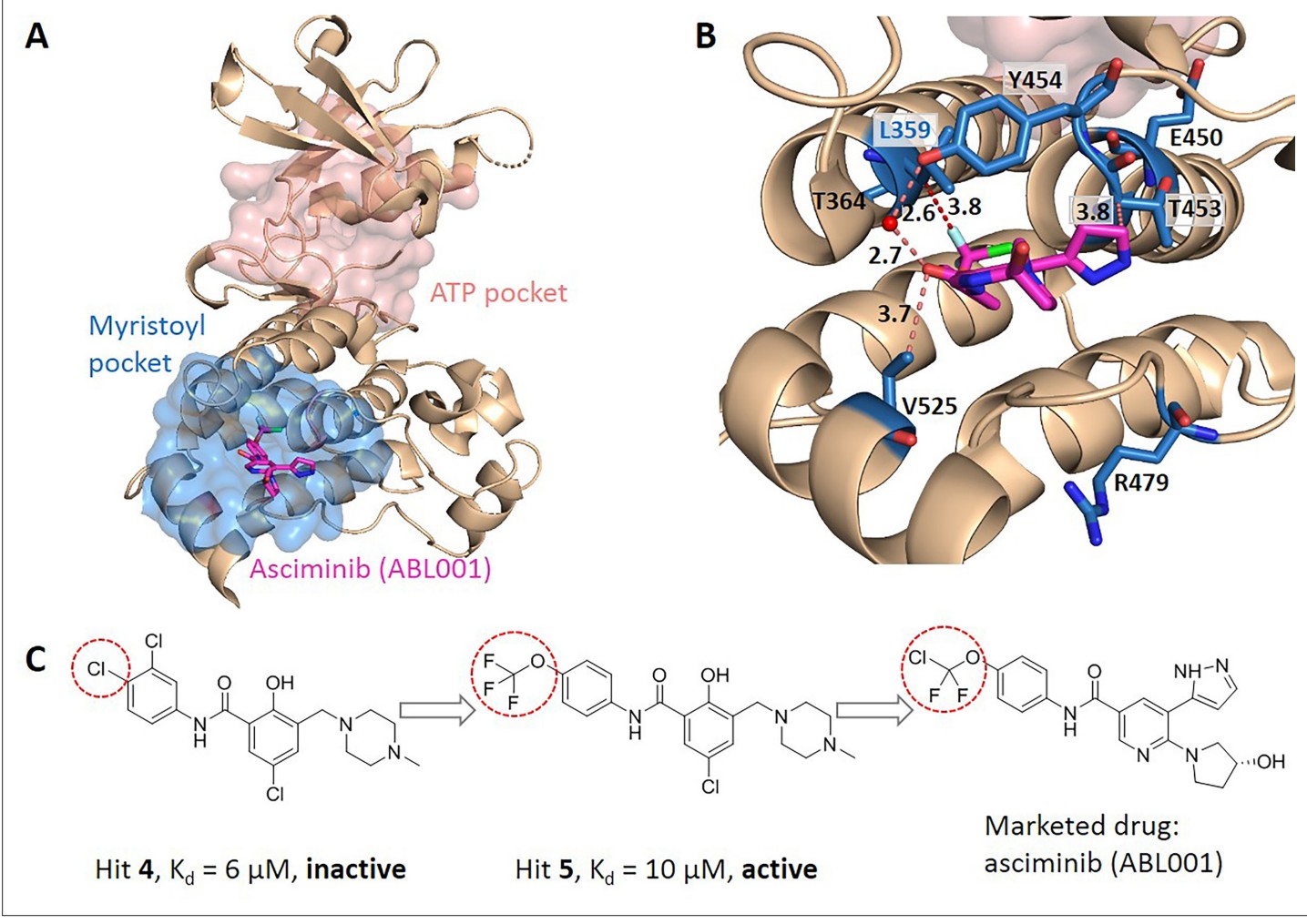

**Figure 4.** Key allo-residues predicted in BCR-ABL1. (**A**) The crystal structure of the kinase domain of BCR-ABL1. The allosteric inhibitor asciminib, represented by sticks, binds to the myristoyl pocket (marine). (**B**) Predicted key allo-residues in the myristoyl pocket. The predicted key allo-residues are represented by marine sticks. One of the predicted key allo-residues, L359, forms a favorable hydrophobic interaction with a fluorine atom in asciminib, represented by a red dashed line. Water is represented by a red sphere. (**C**) The structure of fragment-derived hit **4** and hit **5** and the final marketed drug asciminib.

The online version of this article includes the following source data for figure 4:

**Source data 1.** Raw data for *Figure 4*.

atom at the *para*-position of the aniline to the $CF_3O$- group, hit **5** showed inhibition activity with a slightly weakened $K_d$ of 10 μM compared to hit **4** (*Figure 4C*; *Schoepfer et al., 2018*). This shows that the interaction between $CF_3O$- and BCR-ABL1 is essential for the allosteric signaling and inhibitory activity. This group forms favorable hydrophobic interaction with L359, one of the key allo-residues predicted by our method (*Figure 4B*). In contrast, there is no favorable interaction between hit **4** and L359. These experimental evidences support that the predicted allo-residue L359 plays key role in allosteric signaling.

In the allosteric pocket of the asciminib binding site which contains 44 residues, we predicted 7 key allo-residues. In addition to L359, R479, V525, Y454, E450, T453, and T364 were also identified as potential key allo-residues (*Supplementary file 6*). T453 forms favorable hydrophobic interaction with the pyrazole ring in asciminib, and Y454 participates in the water-mediated H-bond with the oxygen atom in asciminib. Previous studies have shown that the conformational state of helix-I is important for functional activity, and V525 serves as a good indicator for the conformational change. Functional antagonists binding to the myristoyl pocket can bend helix-I and make the disordered region that V525 locates become ordered (*Jahnke et al., 2010*; *Schoepfer et al., 2018*). This indicates that V525

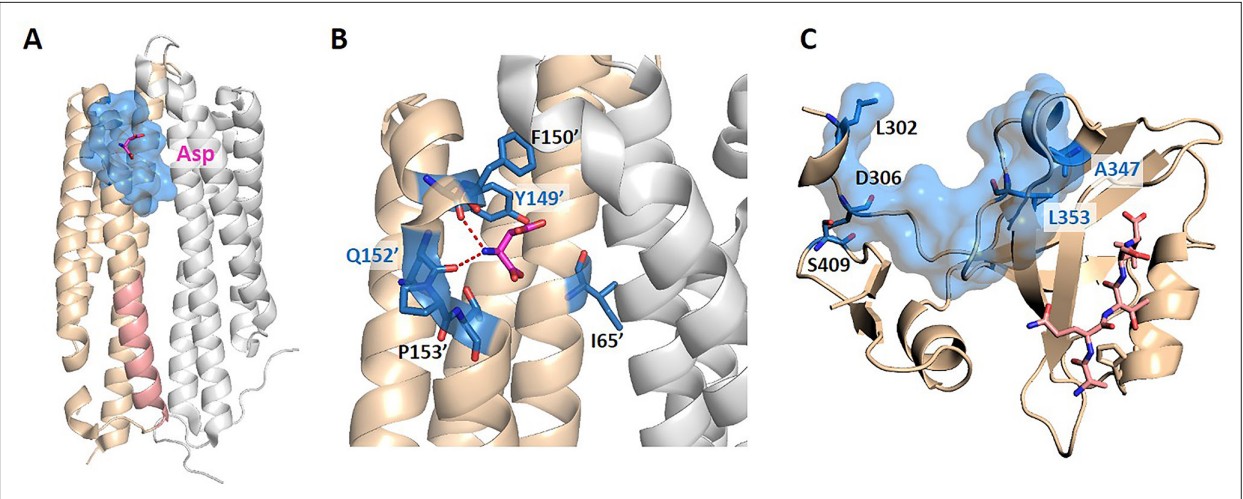

**Figure 5.** The key allo-residues predicted by our method in Tar and PDZ3. (**A**) The crystal structure of holo-Tar. Aspartate (Asp) is represented by magenta sticks, the allosteric pocket is represented by marine surface, and the salmon helix is selected as the orthosteric site. (**B**) The key allo-residues predicted at the Asp-binding site. The predicted key allo-residues in the allosteric cavity_2 are represented by marine sticks, among which Y149 and Q152 are the true key allo-residues that have been confirmed by experiments. Hydrogen bonds are shown as red dash lines. (**C**) The predicted key allo-residues in PDZ3. The peptide bound to the orthosteric site is represented by salmon sticks, the allosteric pocket is represented by marine surface, and the predicted key allo-residues are represented by marine sticks.

The online version of this article includes the following source data for figure 5:

**Source data 1.** Raw data for *Figure 5*.

## KeyAlloSite correctly identified key allo-residues in other proteins not in the data set

We further tested KeyAlloSite on proteins not included in the data set. The first protein is the *E. coli* aspartate (Asp) chemoreceptor Tar. Tar mediates the chemotaxis of bacteria toward attractants, such as Asp, and away from repellents. Tar is a homodimer during transmembrane signaling, and the signals are transmitted from the extracellular region of Tar to the cytoplasmic region through the transmembrane domain (*Mise, 2016*). Bi et al. discovered six new attractants and two new antagonists of Tar by computational virtual screening and experimental study. By comparing the binding patterns of attractants and antagonists, they found that the interactions between the chemoeffectors and Y149 and/or Q152 in Tar are critical for attractant chemotactic signaling (*Bi et al., 2013*). We chose the holo structure that binds Asp (PDB ID: 4Z9H) for analysis and used the CAVITY to identify all potential ligand binding pockets on the surface of chain B. Among the three pockets found, cavity_2 is the pocket where Asp binds, containing 21 residues, which we referred as the allosteric pocket. Since previous studies proposed that transmembrane signaling is triggered by the relative piston-like downward sliding of the α4 helix in the periplasmic domain (*Mise, 2016*), we chose the 16 residues (A166-T181) in the C-terminal of the α4 helix as the orthosteric site (*Figure 5A*). Through the coevolutionary analysis of residues in the orthosteric and allosteric sites, KeyAlloSite identified six key allo-residues in cavity_2, which were I65', F150', Y149', Q152', P153', and T154' (*Figure 5B*). It can be seen that our method could predict the key allo-residues Y149 and Q152. We also tested KeyAlloSite on the apo structure of Tar (PDB ID: 4Z9J). Of the eight pockets found by CAVITY, cavity_5 is the allosteric pocket, which contains 24 residues. As in the holo structure, the 16 residues (A166-T181) in the C-terminal of the α4 helix were chosen as the orthosteric site. In the prediction results, Y149' and Q152' ranked fourth and fifth among the 24 residues, with corresponding Z-scores of 1.01 and 0.66, respectively. Therefore, KeyAlloSite could also correctly predict the key allo-residue Y149 in the apo structure. For Q152, its Z-score is slightly smaller than the threshold of 0.8, though with a high ranking. This indicates

that conformational changes do have subtle influence on the predicted results, probably mainly due to the change of residue composition in the allosteric pocket detected by CAVITY, which will lead to some fluctuations in the predicted key allo-residues. However, when the conformational changes between apo and holo states are not large, the influence on the results is small.

The second protein is the PDZ3 domain, and its allosteric mechanism has been extensively studied. We selected the crystal structure of PDZ3 binding with a peptide in its orthosteric site for analysis (PDB ID: 1BE9; *Doyle et al., 1996*). Since the allosteric site of this structure does not bind an allosteric ligand, we used CAVITY to find the potential ligand binding pockets on its surface. Among the three pockets identified, cavity_1 is the orthosteric pocket, and cavity_2 contains the known allosteric sites, which were used for further analysis. KeyAlloSite predicted D306, S409, L302, A347, and L353 as key allo-residues (*Figure 5C*). Kalescky et al. used rigid residue scan to identify residues that are important for the allosteric effect of the PDZ3 domain. In the rigid residue scan, only one residue was regarded as a single rigid body in each MD simulation. They proposed that A347 is a 'switch residue', which is needed to turn on the allosteric effect (*Kalescky et al., 2015*). Lockless et al. used evolutionary data of protein families to measure the statistical coupling between amino acid positions. For the PDZ protein family, they found that there are strong statistical couplings between A347 and L353 and the key residue H372 of the orthosteric site, and verified using thermodynamic mutational studies (*Lockless and Ranganathan, 1999*). Moreover, McLaughlin et al. developed a high-throughput quantitative method that can individually replace a residue at each position with every other residue for comprehensive single-mutation studies. Their results showed that mutations of A347 and L353 caused significant functional loss (*McLaughlin et al., 2012*). These evidences all indicate that the key allo-residues A347 and L353 we predicted are important for the protein function by allosteric regulation.

## KeyAlloSite identified pathogenetic mutations in human proteins

Previous studies have shown that allosteric mutation, that is, abnormal protein allosteric regulation caused by mutation is related to pathological processes such as cancer (*Kurochkin et al., 2017*). Shen et al. analyzed the dysfunction of allosteric proteins caused by somatic mutations in about 7000 cancer genomes across 33 cancer types, mapped these mutations to allosteric sites, orthosteric sites, and other sites in the Allosteric Database and established the Allo-Mutation data set (*Shen et al., 2017*). We searched for the somatic mutation data corresponding to the human proteins in our data set from the Allo-Mutation data set, and found that 11 of a total of 51 predicted key allo-residues in 7 human proteins were mutated in a variety of cancers (*Table 1*). Among them, cancers that contain a large number of mutations in key allo-residues are uterine corpus endometrial carcinoma and skin cutaneous melanoma. This indicates that the abnormal allosteric regulation caused by the mutation of key allo-residues plays key role in the occurrence and development of cancer. These key allo-residues can affect allosteric signal transduction and thus affect protein functions, suggesting that KeyAlloSite can be used to predict key pathogenetic mutations in proteins.

**Table 1.** Predicted key allo-residues that were mutated in cancers.

| Protein | Gene | Predicted key allo-residues | Mutation* | Cancer type† |
|---------|------|-----------------------------|-----------|--------------|
| AR1 | AR | D732 | D732N | SKCM |
| AR2 | AR | M832 | M832I | SKCM |
| PTP-1B | PTPN1 | M282 | M282T | COAD |
| CDK2 | CDK2 | P155 | P155H | UCEC |
| CK2alpha | CSNK2A1 | F54; A110 | F54C; A110T | UCEC; UCEC, GBM |
| MAPK14 | MAPK14 | P191; E192 | P191S; P191H; E192Q | SKCM; KIRC; BLCA |
| MAPK8 | MAPK8 | E195; M200 | E195K; M200I | UCEC; SKCM |
| CYP3A4 | CYP3A4 | F219 | F219L | UCEC |

*Mutation: confirmed disease mutations among the predicted key allo-residues.

†Cancer type: COAD: colon adenocarcinoma; SKCM: skin cutaneous melanoma; UCEC: uterine corpus endometrial carcinoma; GBM: glioblastoma multiforme; KIRC: kidney renal clear cell carcinoma; BLCA: bladder urothelial carcinoma.

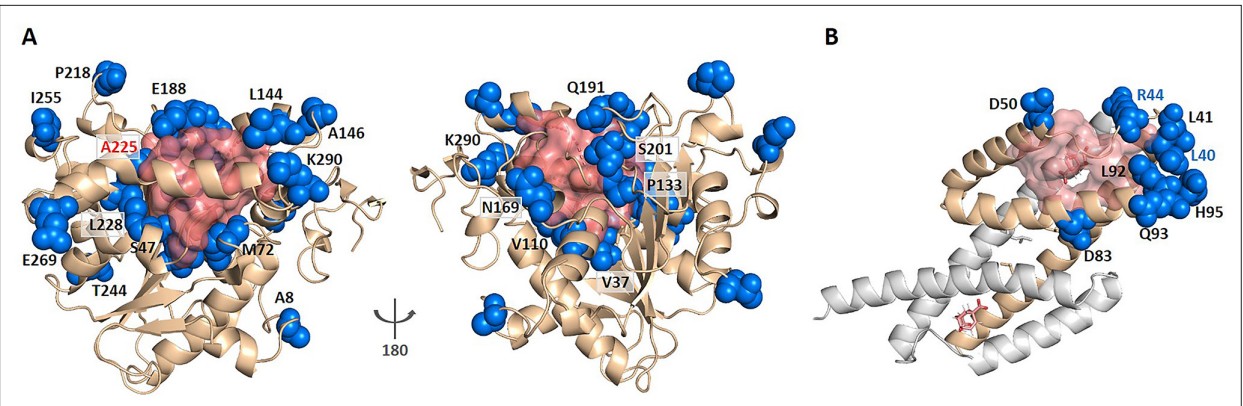

**Figure 6.** KeyAlloSite predicted key allo-residues for enzymes. (**A**) KeyAlloSite predicted key allo-residues for *Candida antarctica* lipase B. Among the predicted residues, the residues that have been annotated by the literature are shown as marine spheres, and the orthosteric pocket is represented by salmon surface. (**B**) KeyAlloSite predicted key allo-residues for *Escherichia coli* chorismate mutase (CM). Experimentally discovered key functional residues of CM are shown as marine spheres, the labels of key allo-residues predicted by KeyAlloSite are shown in marine, and the orthosteric pocket and ligand are represented by salmon surface and sticks.

The online version of this article includes the following source data for figure 6:

**Source data 1.** Raw data for *Figure 6*.

## KeyAlloSite can also identify key allosteric functional residues of enzymes

Enzyme evolution studies mainly focus on mutations in the substrate binding pocket (*Wu et al., 2013*; *Yi et al., 2011*). However, covariant residues far from the substrate binding site may also play an important role in regulating catalysis, which are difficult to identify. Since KeyAlloSite can calculate the functional coupling between residues outside and inside the orthosteric pocket, we wonder whether it can also be used to identify key allo-residues for enzyme activity regulation. We used *Candida antarctica* lipase B (CALB, PDB ID: 1TCA), which has many annotated key functional residues (*Wu et al., 2020*), as a test case. CALB is one of the most widely used biocatalysts in academia and industry that is often applied in acylating kinetic resolution of racemic alcohols and amines and desymmetrization of diols and diacetates, and is robust and easy to express (*Wu et al., 2013*). We selected the six angstrom-cutoff orthosteric pocket and calculated the EC values between each residue outside the orthosteric pocket and the residues in the orthosteric pocket. Then we compared the difference in EC values corresponding to each residue outside the orthosteric pocket and calculated the significant difference number $d_i$ corresponding to each residue. Finally, the significant difference number $d_i$ was normalized to Z-score. Due to the large number of residues outside the orthosteric pocket, residues with Z-scores greater than 0.9 were referred as key allo-residues here. Our method predicted a total of 52 residues from the 296 residues outside the orthosteric pocket, of which 20 residues have been annotated as functional residues in literature according to mutagenesis experiments (*Figure 6A*, *Supplementary file 7*). For example, A225 ranked the third out of the 52 predicted residues with a Z-score of 3.04, and A225M improves the catalytic efficiency of the enzyme by about 11 folds. V37 ranked the 35th out of the 52 predicted residues with a Z-score of 1.42, and V37I improves the catalytic efficiency of the enzyme by about threefolds (*Wu et al., 2020*). Therefore, our method can also be used to predict allosteric functional residues that are important for enzyme catalysis, providing a new computational tool for identifying mutant enzyme with improved catalytic properties.

Russ et al. recently used an evolution-based model to design chorismate mutase enzymes (CMs). They used DCA to learn the constraints for specifying proteins purely from evolutionary sequence data and performed Monte Carlo sampling from this model to generate artificial sequences. They were able to obtain proteins with natural-like catalytic function with sequence diversity. Eight residues (L40, L41, R44, D50, D83, L92, Q93, and H95) at the periphery of the active site were found to be important for CM catalysis in *E. coli* specific function (*Russ et al., 2020*; *Figure 6B*). We wonder if some of these residues regulate the enzyme function by allostery. We used the 6 angstrom-cutoff orthosteric pocket and identified 46 residues on the surface of the protein outside the orthosteric

pocket. We calculated the EC values between each of these 46 residues and the orthosteric pocket and compared the difference in EC values of these 46 residues. We then calculated the significant difference number $d_i$ corresponding to each of these 46 residues and normalized to Z-scores. Using the criterion of Z-score >0.9, we predicted 10 key allo-residues from the 48 residues, which were L86, A9, R44, Q89, E57, E12, L40, D69, R58, and N5. Among them, two residues (R44 and L40) were shown to be important for CM catalysis by Russ et al. And Russ et al. used DCA to show that sequence-based statistical models contain protein function information, while our study used DCA to predict allosteric sites or key allosteric residues based on the significance analysis of EC between these sites with the orthosteric site. Their study showed that the function of proteins seems to depend on many weak terms in $J_{ij}$, which lack simple physical interpretations. Our study demonstrated that weak terms in $J_{ij}$ contain information for protein allosteric function evolution and can be used to predict allosteric sites and key allosteric residues.

To further verify the effectiveness of KeyAlloSite, we made indirect comparisons with the statistical coupling analysis (SCA) method. SCA predicts several groups of coevolved residues (sectors) that form physically continuous networks, often able to connect major functional site and allosteric sites, that is, allosteric pathways (*Lockless and Ranganathan, 1999*; *Rivoire et al., 2016*; *Salinas and Ranganathan, 2018*; *Shulman et al., 2004*; *Süel et al., 2003*). We analyzed all the proteins in our data set using SCA. We first compared the performance of SCA and KeyAlloSite in predicting known key allo-residues in allosteric sites (*Supplementary file 9*). For the known key allo-residue L359 in the BCR-ABL1 protein, it was not present in the sectors predicted by SCA, despite that the sectors contain 68 residues, while it could be correctly predicted by KeyAlloSite. For the known key allo-residues Y149 and Q152 in the Tar receptor, KeyAlloSite could correctly predict both of them. However, although the sectors predicted by SCA contained Y149 and Q152, it also included two residues (R69 and R73) that have been experimentally verified to contribute only to ligand binding and not to allosteric signaling (*Bi et al., 2013*). For the PDZ3, the sectors predicted by SCA contained the key allo-residues A347 and L353, which were also successfully predicted by KeyAlloSite. We further compared the performance of SCA and KeyAlloSite in predicting the key allosteric functional residues of enzymes. For the CALB, the sectors predicted by SCA missed one of the key allo-residues A225, which has been experimentally shown to have a great impact on enzyme activity, and the predicted known key allo-residues account for 32.8% of all the residues in the sectors, while KeyAlloSite could predict the key allo-residue A225, and the predicted known key allo-residues account for 38.5% of all the predicted key allo-residues. For the CMs, the sectors predicted by SCA contained only one key allo-residue D83, while KeyAlloSite could predict key allo-residues R44 and L40. For the KeyAlloSite correctly predicted functional phosphorylation sites T288 and T287 in the AurA, SCA missed both of them. Thus, KeyAlloSite performs better than SCA in predicting key allo-residues.

## Discussion

Identifying key allosteric residues responsible for allosteric signaling is important for the design and optimization of allosteric drugs, enzyme, and protein engineering studies. We developed, KeyAlloSite, a novel method for predicting allosteric sites and key allo-residues based on the ECM. To the best of our knowledge, this is the first systematic and efficient computational method to predict key allo-residues. Our study demonstrated that orthosteric and allosteric pockets are coupled in protein function evolution. Our predicted key allo-residues are in accordance with previously reported experimental studies in the BCR-ABL1, Tar, and PDZ3 systems, as well as key cancer mutations. We further showed that KeyAlloSite can be applied to predict key allosteric residues distant from the catalytic site that are important for enzyme catalysis. Our study also gives a possible physical explanation for the weak couplings in $J_{ij}$, that is, they may represent allosteric functional couplings. The predicted key allo-residues can help us to understand the mechanism of allosteric regulation, to provide reference and guidance for the rational design and optimization of allosteric drugs and to facilitate enzyme engineering.

It should be noted that due to limited available experimental information, for the protein systems that we tested, although all the predicted known key allo-residues ranked among top 20%, there are other residues among the top list with unidentified function. Our analysis predicted that these residues should play important roles in allosteric signaling. Further experimental studies are needed to verify their functions in the future. At the same time, our predicted list of key allo-residues greatly

reduces the number of residues that needs to be verified experimentally. We expect future experimental studies on the functions of these newly predicted key 'allo-residues' can not only verify and demonstrate the predictive power of KeyAlloSite but also offer more data to improve it.

Several methods were developed to infer protein function based on the coevolutionary couplings between residues. For example, Hopf et al. developed EV mutation to predict the effects of mutations based on the coevolutionary couplings between residues (*Hopf et al., 2017*). The theoretical formulation for extracting the coupling scores between residues in the initial step of our method is the same as that of Hopf et al., but the problem studied by us is different from that of Hopf et al., and the usage of the coupling scores between residues in the later steps of the two methods is different. Hopf et al. used the coevolutionary coupling scores between residues to predict the effects of mutations by calculating the difference in statistical energy between mutant and wild-type sequences. In contrast, we used the coevolutionary coupling scores between residues to predict the allosteric sites and key allo-residues in allosteric pockets that are mainly responsible for allosteric signaling by pairwise comparing the difference of the coevolutionary coupling scores of residues in allosteric pockets. Although Hopf et al. highlighted the significance of the pairwise coupling term for the prediction of mutation effects, we highlighted the importance of the weak pairwise coupling term for the study of allosteric function.

As KeyAlloSite attempts to capture coevolutionary coupling between residues from MSA, it requires that MSA should contain sufficient homologous and diversified sequences. For the MSAs with only a few homologous sequences, KeyAlloSite usually cannot give accurate predictions. How to reduce the number of homologous sequences required remains further research.

The scores of the key allo-residues predicted by KeyAlloSite depend not only on the coevolutionary information but also on the residues contained in allosteric pockets. When different conformations of the protein are used, pocket detection method may give allosteric sites with slightly or largely different residues (depending on the difference of the conformations) that may influence the final KeyAlloSite. For example, in the case of Tar, the ECs between residues are the same for the apo and holo conformations, while the allosteric pockets found by CAVITY in the two conformations contained a small number of different residues. Because the prediction of key allo-residues by KeyAlloSite requires pairwise comparison of residues in allosteric pockets, the predicted key allo-residues in the two conformations were slightly different. For the apo Tar, on the one hand, although the score of Q152 is 0.66, which is less than the threshold of 0.8, Q152 ranked high among all residues in the allosteric pocket with a ranking of 5/24. When we lower the threshold slightly, we will be able to correctly predict Q152. On the other hand, Bi et al. showed that although Y149 and Q152 are both key allo-residues, Y149 seems to be more important as allosteric signaling can be conducted when the allosteric molecule only interacts with Y149 but not Q152 (*Bi et al., 2013*). Although the holo Tar has some conformational changes compared to apo Tar, the key allo-residueY149 can be captured by KeyAlloSite when using either the holo or the apo structure. For applications, we recommend to use the holo structure whenever possible.

Although we used the three-dimensional structure of proteins and their binding ligands in our analysis, KeyAlloSite can also be applied in cases where no three-dimensional structures are available on condition that a certain number of homologous sequences of the protein under investigation and location of the functional site are known. Along with the rapid progress in recent years, protein structure prediction methods, such as AlphaFold (*Jumper et al., 2021*), can be used to predict the protein structure first. At the same time, as our method calculates the EC between any residue of the protein and residues of the orthosteric pocket, KeyAlloSite can be used to predict not only key allosteric residues but also PTM sites that have functional correlation with orthosteric sites, which will be further studied in the future.

Key residues in protein allosteric sites determine the direction and strength of allosteric signaling, while other residues in allosteric sites mainly contribute to binding. When optimizing allosteric molecules, we often face the challenge that simply increasing binding affinity cannot improve the efficacy of allosteric regulation. Therefore, to optimize the allosteric molecules, one can first use KeyAlloSite to predict the key allo-residues in the allosteric pocket and then maintain or enhance the interactions between the allosteric molecules and the key allo-residues while altering the interactions between the allosteric molecules and other residues in the allosteric pocket. In this way, the optimized allosteric molecules can be ensured to have both strong binding affinity and allosteric signal transduction ability.

## Materials and methods

### The allosteric protein data set

We selected allosteric proteins from the 'Core Set' of ASBench (*Huang et al., 2015*) and constructed our data set according to the following criteria: (1) the protein functions as a monomer; (2) the corresponding three-dimensional protein structure data should contain both allosteric ligand and orthosteric ligand; (3) the number of effective homologous sequences of the protein should be greater than 100. Finally, 23 allosteric proteins were selected, including 25 known allosteric sites (*Supplementary file 1*). We collected another two proteins from published literatures, including *E. coli* aspartate chemoreceptor Tar (PDB ID: 4Z9H) (*Mise, 2016*) and PDZ3 domain (PDB ID: 1BE9) (*Doyle et al., 1996*), as key allo-residues have been reported for these two proteins that can be used for comparative analysis. We used HMMER to search for the homologous sequences of each of the selected allosteric proteins from pfam (*Finn et al., 2015*). Due to the redundancy of homologous sequences, they were reweighted according to the standard of 80% sequence identity to obtain the effective homologous sequences.

### The evolutionary coupling model

We used a global statistical model, the ECM, which can calculate the direct couplings between residues and remove the indirect couplings. The ECM we used here was mainly based on the work of Marks and her co-workers (*Ekeberg et al., 2013*; *Weinreb et al., 2016*). From the MSA of a protein family, we can calculate the observed frequency $f_i^a$ and pairwise co-occurrences $f_{ij}^{ab}$ of residues ($a$, $b$) at position ($i$, $j$). From this first-order and second-order statistics, we can infer a model to account for the observed statistics optimally, which mainly includes two parameters: single-site propensities $h_i(a)$ and direct coevolutionary couplings between residues $J_{ij}(a,b)$. This model defines a probability $P$ for each protein sequence $\boldsymbol{a} = (a_1, \ldots, a_L)$ of length $L$:

$$P(\boldsymbol{a}) = \frac{1}{Z} \exp\left( \sum_{i=1}^{L} h_i(a_i) + \sum_{i=1}^{L-1} \sum_{j=i+1}^{L} J_{ij}(a_i, a_j) \right) \tag{1}$$

$$Z = \sum_a \exp\left( \sum_{i=1}^{L} h_i(a_i) + \sum_{i=1}^{L-1} \sum_{j=i+1}^{L} J_{ij}(a_i, a_j) \right) \tag{2}$$

The parameters $\boldsymbol{h}$ and $\boldsymbol{J}$ of the model were estimated by pseudo-maximum likelihood (PLM) which approximates the full likelihood for each sequence $\boldsymbol{a} = (a_1, \ldots, a_L)$ by a product of conditional likelihoods for each site $i$:

$$P(a_1, \ldots, a_L | \boldsymbol{h}, \boldsymbol{J}) \approx \prod_{i=1}^{L} P(a_i | \boldsymbol{a} \backslash a_i, \boldsymbol{h}, \boldsymbol{J}) \tag{3}$$

The global partition function Z is replaced by a number of local partition functions:

$$P(a_i | \boldsymbol{a} \backslash a_i, \boldsymbol{h}, \boldsymbol{J}) = \frac{\exp\left( h_i(a_i) + \sum_{j \neq i} J_{ij}(a_i, a_j) \right)}{\sum_a \exp\left( h_i(a) + \sum_{j \neq i} J_{ij}(a, a_j) \right)} \tag{4}$$

After modifying with $L_2$-regularization and sample reweighting, the approximate likelihood function was optimized using a quasi-Newton method (Limited-memory BFGS).

Once the parameters $\boldsymbol{h}$ and $\boldsymbol{J}$ are fitted, the Frobenius norm $FN(i,j)$ of $J_{ij}$ is used to measure the ECS between the two sites $i$ and $j$.

$$FN(i,j) = \|J_{ij}\|_2 = \sqrt{\sum_k \sum_l J_{ij}(k,l)^2} \tag{5}$$

$J'_{ij}$ was obtained after $J_{ij}$ was centralized.

Since the $FN$ may include bias caused by phylogeny and undersampling, it can be corrected with average product correction (*Dunn et al., 2008*). The EC value is the EC score of the two sites.

$$EC(i,j) = FN(i,j) - \frac{\left( \sum_{i' \neq i} FN(i',j) \right) \left( \sum_{j' \neq j} FN(i,j') \right)}{\sum_{i'} \sum_{j' \neq i'} FN(i',j')} \tag{6}$$

## Evolutionary coupling strength between orthosteric and other pockets

We first performed MSA (*Figure 1A*) and calculated the ECs between residues using the ECM (*Figure 1B*). We then used the CAVITY program (*Xu et al., 2018*; *Yuan et al., 2013*) to identify all the potential ligand binding pockets on the surface of a protein and designated the $m$th pocket as cavity_m. The ECS ($ECS_{\text{cavity}\_m}$ between the orthosteric pocket and the $m$th pocket is defined as the sum of the coupling strength between the residues in the two pockets. The orthosteric pocket here is defined as all of the residues within 6 Å around the orthosteric ligand. If more than 50% of the residues in the $m$th pocket overlap with the orthosteric pocket, the pocket will be excluded. Otherwise, the overlapping residues between the $m$th pocket and orthosteric pocket will be removed from the $m$th pocket, and the remaining residues will be used to calculate the ECS between the $m$th pocket and orthosteric pocket.

$$ECS_{\text{cavity}\_m} = \sum_{i\in\text{cavity}\_m} \sum_{j\in\text{orthosteric pocket}} FN\left(i,j\right) \tag{7}$$

Finally, we normalized the ECS of the pockets:

$$Z\text{-}score_{\text{cavity}\_m} = \frac{ECS_{\text{cavity}\_m} - \mu_{\text{cavity}}}{\sigma_{\text{cavity}}} \tag{8}$$

$\mu_{\text{cavity}}$ is the mean value of the ECS between the orthosteric pocket and the other pockets, and $\sigma_{\text{cavity}}$ is the standard deviation of the ECS.

## Identification of key allo-residues

We first calculated the pairwise EC value between one residue $a_i$ ($i=1,2,...,N$) in the allosteric pocket and one residue $b_j$ ($j=1,2,...,M$) in the orthosteric pocket. An $N \times M$ matrix $\boldsymbol{E}$ was obtained, and each element $E_{ij}$ in the matrix $\boldsymbol{E}$ represents the EC value between residues in the allosteric and orthosteric pockets (*Figure 1C–D*). The allosteric pocket here refers to the allosteric pocket found by the CAVITY. In rare cases, if the CAVITY does not find the known allosteric pocket, all of the residues within 8 Å around the allosteric ligand are used as the allosteric pocket. After that, the difference of EC values corresponding to each residue in allosteric pocket was pairwisely compared by using the student's t-test ($\alpha=0.05$), that is, whether there was a difference between the mean values of each two rows of matrix $\boldsymbol{E}$. The result of the comparison was expressed by $C_{mi}$ ($m=1,2,...,N$, $i=1,2,...,N$), if there was a significant difference, $C_{mi}$ was assigned a value of 1, if there was no significant difference, $C_{mi}$ was assigned a value of 0. Finally, an $N \times N$ matrix $\boldsymbol{C}$ was obtained, and each element in $\boldsymbol{C}$ indicates whether there was a difference between the residues in the allosteric pocket (*Figure 1E*). On this basis, by adding up each column of $\boldsymbol{C}$, we could get the number of significant differences $d_i$ ($i=1, 2,..., N$) between each residue and the remaining residues in the allosteric pocket, that is, the number of significant differences corresponding to $i$th residue (*Figure 1F*).

$$d_i = \sum_{m=1}^{N} C_{mi} \tag{9}$$

Finally, we normalized the number of significant differences of the residues:

$$Z\text{-}score_{\text{residue}\_i} = \frac{d_i - \mu_{\text{residue}}}{\sigma_{\text{residue}}} \tag{10}$$

$\mu_{\text{residue}}$ is the mean value of the number of significant differences of the residues, and $\sigma_{\text{residue}}$ is the standard deviation of the number of significant differences.

In enzyme catalysis, in addition to residues in the active site, distant residues may also affect enzyme activity, which could not be rationally designed in enzyme engineering. We systematically studied the effect of residues outside of the active site on enzyme catalysis. We calculated the EC value between each residue outside the orthosteric pocket and residues in the orthosteric pocket (i.e. the rows in matrix $\boldsymbol{E}$ are all residues outside the orthosteric pocket here). Using the same steps as above, we determined the number of significant differences $d_i$ for each residue outside the orthosteric pocket and normalized it to Z-score.

## Data availability

The data that support the results of this study are in the Supplementary data, including information of the allosteric proteins in the data set (*Supplementary file 1*); list of the Z-scores and ranking of allosteric pockets in the data set (*Supplementary file 2*); KeyAlloSite prediction results of AurA kinase (*Supplementary file 3*); list of the predicted key allo-residues in allosteric pockets (*Supplementary file 4*); key allo-residues predicted by KeyAlloSite with different cutoffs (*Supplementary file 5*); KeyAlloSite prediction results of tyrosine-protein kinase ABL1 (*Supplementary file 6*); the key allo-residues predicted by our method on CALB (*Supplementary file 7*); the confusion matrices of KeyAlloSite in different scenarios (*Supplementary file 8*); comparison of KeyAlloSite and SCA methods (*Supplementary file 9*); phylogenetic tree of androgen receptor (*Figure 2—figure supplement 1*); comparison of ECS between pockets when all residue pairs and partial residue pairs were used (*Figure 2—figure supplement 2*); difference between the EC between orthosteric and allosteric sites and the EC between two random patches (*Figure 2—figure supplement 3*); distribution of the ratios of the number of key allo-residues predicted by KeyAlloSite in the number of all residues in allosteric pockets when using different cutoffs in all proteins (*Figure 3—figure supplement 1*); examples of distributions of the statistics corresponding to significant scores obtained from the t-test (*Figure 3—figure supplement 2*); and random sampling of homologous sequences (*Figure 3—figure supplement 3*). The homologous sequences of the proteins in the data set are available in the following GitHub repository: https://github.com/huilan1210/KeyAlloSite, *Xie et al., 2023*.

## Code availability

KeyAlloSite is available at GitHub (https://github.com/huilan1210/KeyAlloSite, copy archived at swh:1:rev:8464b27b588af48d14033ab40d62f9eca4ed0051, *Xie et al., 2023*).

## Acknowledgements

The authors thank Jintao Zhu from the Center for Quantitative Biology and Gaoxiang Pan from the College of Chemistry and Molecular Engineering, Peking University for helpful discussions. This study was supported in part by the National Key R&D Program of China (2022YFA1303700), the National Natural Science Foundation of China (21633001, 22237002) and the Chinese Academy of Medical Sciences (2021-I2M-5–014).

## Additional information

### Funding

| Funder | Grant reference number | Author |
| --- | --- | --- |
| National Key R&D Program of China | 2022YFA1303700 | Luhua Lai |
| National Natural Science Foundation of China | 21633001 | Luhua Lai |
| Chinese Academy of Medical Sciences | 2021-I2M-5-014 | Luhua Lai |
| National Natural Science Foundation of China | 22237002 | Luhua Lai |

The funders had no role in study design, data collection and interpretation, or the decision to submit the work for publication.

### Author contributions

Juan Xie, Conceptualization, Data curation, Formal analysis, Investigation, Visualization, Methodology, Writing - original draft, Writing - review and editing; Weilin Zhang, Conceptualization, Formal analysis, Investigation, Methodology; Xiaolei Zhu, Methodology, Writing - review and editing; Minghua Deng, Conceptualization, Formal analysis, Methodology; Luhua Lai, Conceptualization, Supervision, Funding acquisition, Methodology, Writing - review and editing

**Author ORCIDs**
Juan Xie http://orcid.org/0000-0001-6975-0449
Luhua Lai http://orcid.org/0000-0002-8343-7587

**Decision letter and Author response**
Decision letter https://doi.org/10.7554/eLife.81850.sa1
Author response https://doi.org/10.7554/eLife.81850.sa2

## Additional files

### Supplementary files
- Supplementary file 1. Information of the allosteric proteins in the data set.
- Supplementary file 2. List of the Z-scores and ranking of allosteric pockets in the data set.
- Supplementary file 3. KeyAlloSite prediction results of Aurora A kinase.
- Supplementary file 4. List of the predicted key allo-residues in allosteric pockets.
- Supplementary file 5. Key allo-residues predicted by KeyAlloSite with different cutoffs.
- Supplementary file 6. KeyAlloSite prediction results of tyrosine-protein kinase ABL1.
- Supplementary file 7. The key allo-residues predicted by our method on *Candida antarctica* lipase B.
- Supplementary file 8. The confusion matrices of KeyAlloSite in different scenarios.
- Supplementary file 9. Comparison of KeyAlloSite and SCA methods.
- MDAR checklist

### Data availability
All data that support the results of this study are included in the manuscript, supplementary files, and GitHub repository (https://github.com/huilan1210/KeyAlloSite; copy archived at swh:1:rev:-333d4a48d2570c74806f68a1247611ed64794b97). Source Data files have been provided for all Figures (except Figure 1 and Figure 2-figure supplement 1).

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
