## [Editor Report]

The manuscript reports on a useful tool to study protein allosteric regulation function. The work is based on inadequate experimental validation of the predicted residues implicated in mediating allosteric signaling. The study highlights the significance of the weak pairwise term for the prediction of the allosteric function.

---

## [Decision Letter]

**Decision letter after peer review:**

Thank you for submitting your article "Coevolution-based prediction of key allosteric residues for protein function regulation" for consideration by *eLife*. Your article has been reviewed by 2 peer reviewers, and the evaluation has been overseen by a Reviewing Editor and Volker Dötsch as the Senior Editor. The following individual involved in review of your submission has agreed to reveal their identity: Sarath Dantu (Reviewer #2).

Essential revisions:

There seems to be a consensus between the reviewers that:

1) There is a lack of comparison of the proposed method against existing ones that do pretty much the same. This is the main weakness of the manuscript.

2) The authors should also provide a better metric of the true positives of their method – and of course report the false positives they have for the cases they've tested. AlloSite seems to predict several residues as key, some of which happen to be oncogenic (Table 1). It's not clear how many of the top 10, let's say, residues that it predicts have an allosteric effect.

3) In addition to the comparison, the authors need to establish that the signal coming from comparison of allosteric vs. orthosteric is different from comparing any two random patches with same number of residue pairs.

4) A complete picture of the predictive power with both strengths and weakness of the tool should be presented.

*Reviewer #1 (Public Review):*

Allostery refers to processes whereby a change at one site of a biological macromolecule affects the structure and dynamics at another distinct functional site, enabling the regulation of the corresponding function. Xie et al. developed an in-silico method to predict residues involved in allosteric regulation using a coevolution-based method. A fast and accurate method of identifying key residues responsible for allosteric signalling is important for drug design purposes and protein engineering.

Strengths:

1) The authors applied their method to multiple targets from different protein families to test their method.

2) The method is able to predict in a retrospective analysis certain residues involved in allosteric communication between orthosteric and allosteric binding sites.

Weaknesses:

1) There are several tools used in statistical genomics to predict allosteric communication pathways. Even though the paper tries to demonstrate the ability to predict residues that are involved in allosteric communication, KeyAllosite is not compared with any other state-of-the-art tool that does the same (1-3), which would highlight the strengths of this method with respect to existing ones.

2) The authors mention that the number of effective homologue sequences affects the probability of finding an allosteric site in the top 3 scored sites, however, Cdc4 and AR2 have also a low number of effective homologue sequences (Figure 1.A) yet a high Z-score. No sufficient explanation is given regarding this discrepancy. This also demonstrates that a threshold in the Neff under which KeyAlloSite becomes unreliable should be defined.

3) From the low Z-score of CYP3A4, the authors claim that the conservation of residues in the orthosteric site is important for getting an accurate prediction of the coupling strength. It is not clear though if and how is this aspect encoded in the KeyAlloSite. From the description of the method, it seems like the algorithm does not check for the level of conservation of the orthosteric residues. An explanation as to why has it not been incorporated into the algorithm is necessary. The conservation at a given site in an MSA defined as the overall deviance of amino acid frequencies at that site from their mean values, in combination with the statistical coupling of two sites has been shown to be important in the development of allosteric models (4).

4) In the case of AuroraA, the authors do not explain why other Ser/Thr/Tyr residues scored higher than T287 and T288, or if their higher scores are an artefact. However, in many cases, post-translational modifications take place by secondary partners and, therefore, the coupling of a post-translational modification site with the orthosteric site cannot be used to predict such sites. Post-translational modification sites are expected to have a strong coupling with residues of the upstream/downstream effector that is responsible for the modification, rather than with residues of the orthosteric site of the protein.

5) The authors defined allo-residues as the residues whose Z-score is >0.8, but there is no strong argument regarding the choice of this threshold. In the case of the allosteric pockets, for example, the authors use a threshold of 0 to identify pockets with strong coupling strength.

6) In the case of the Tar, it would be good if the authors reported the results starting from an apo structure as well and see if the method is able to find Y149 and Q152. That way, they could test how sensitive/biased the method is to the chosen conformation. If no apo structure is available, maybe another protein where both an apo and a holo structure exist could be used for this purpose.

7) The data in Table 1 is not sufficiently convincing. It would be helpful to also report how many of the identified residues by KeyAlloSite are indeed involved in oncogenic mutations or find another metric to quantify the success rate of KeyAlloSite.

8) The success rate of the method to predict key residues of the function of CALB (38% success rate based on the reported residues in the literature) and CMS (20%), should be interpreted with caution. It may be the case that the rest of the predicted residues have not been tested for their functional role, or that the method has indeed a low success rate in predicting key residues for allosteric communication.

In lines 38-41, the authors refer to SARs of allosteric drugs as "flat", which is not clear what they refer to.

It is not clear if the so-called, "allo-residues", that the authors define in lines 49-52 refer to residues that affect the binding or the signalling.

It would be helpful for the reader if the authors included a section where they describe the method itself and the key steps of developing their model before presenting the results. That way, the reader could follow the results easier.

The authors classified L359 of BCR-ABL1 as an allo-residue and justified the importance of this residue based on the fact that a weaker ligand does interact with this residue. For transparency, it would be good for the authors to report the allosteric scores of all 44 residues in the allosteric site to show that this residue was not cherry-picked and that the method can pick up all the residues that are important for the binding.

The manuscript has several typos and language mistakes (e.g. "screening" instead of "screen" in the Abstract, missing articles, etc.) that should be corrected prior to a resubmission.

*Reviewer #2 (Public Review):*

The authors designed a statistical approach to analyse coevolution scores from a protein and predict allosteric residues. This approach relies on comparison of residues from the two sites (allosteric vs. orthosteric) and omits the rest of the protein.

While the approach is logical, the predictive power has not been clearly established. To demonstrate the effectiveness of the approach a confusion matrix should be provided as it will show the predictive power of the method in the different scenarios presented in the manuscript (PTM's, enzymes, pathogenic mutations, etc.).

An effective method, along these lines, will have significant impact on protein and drug design.

It would be helpful if statistics on the distribution of significant scores from E row comparison after the t-test are provided. Further, you have so far compared allosteric with orthosteric, it would be interesting to compare the same distribution/statistics with allosteric vs. non-orthosteric residues. It may serve as a very good benchmark. The idea would be to see if the events of significance from allosteric vs. orthosteric are different/unique if we pick and compare any two random patches in the protein.

A boxplot of Zscores for All, top200, top300, top400 pairs from Figure 1—figure supplement 2 would be very informative.

As you have been using top zscore residues (>0.5 or >0.9 for enzymes), direct or indirect evidence for the statement in lines 306-307 is not very apparent from the provided data. This has to be flushed out further.

Please annotate residues for which experimental data is available to compare with keyallosite predictions for example in Supplementary files 4 and 5. In the main text (line 284) you only mention one residue out of 52 to demonstrate the effectiveness of the prediction. Again a confusion matrix would help here.

Can you please clarify if in Figure 2, number of residues refers to the number of residues from allosteric+orthosteric site?

Figure 6 may be ideal as Figure 1 to inform the readers about the protocol of this work.

[Editors' note: further revisions were suggested prior to acceptance, as described below.]

Thank you for resubmitting your work entitled "Coevolution-based prediction of key allosteric residues for protein function regulation" for further consideration by *eLife*. Your revised article has been evaluated by Volker Dötsch (Senior Editor) and a Reviewing Editor.

The manuscript has been improved but there are some remaining issues that need to be addressed, as outlined below:

1. Can you please provide a detailed description and expand the discussion on the predicted amino acids and why the predicted residues are not the top ranking ones?

2. The theoretical formulation to extract the coupling score and a compare/contrast to Hopf et al. approach would be highly beneficial to the readers.

*Reviewer #1 (Recommendations for the authors):*

The authors did extensive work trying to address the revision comments, and their effort is much appreciated.

Nevertheless, in the absence of further experimental validation of the importance of the predicted so-called key "allo-residues" in mediating the signal between the orthosteric and allosteric binding site, it is still hard to assess the predictive power of the presented method. There are still residues for example that score higher than known functional residues (see for example residues T235, S245 and S249 in the case of Aurora A) whose implication in signal transduction is not confirmed, or residues whose importance depends on the conformation chosen for analysis despite the coevolutionary conservation metric used to score the residues (Q152 of Tar whose score is way below the 0.8 threshold when an apo structure is considered, which shouldn't be the case given that AlloSite uses only MSA and coevolutionary information for the scoring – it seems like other residues are predicted to be way more important based on the allosteric pocket definition given by CAVITY to decrease the score of Q152 to <0.8 after normalisation).

I fully understand that such experimental validation goes beyond the capacities of a computational group, however, as a reader, I might be hesitant to try this method.

*Reviewer #2 (Recommendations for the authors):*

The authors have carefully addressed all the comments from the previous review.

Few additional comments for authors to consider:

The Discussion section can be enriched further. At present it only discusses two points, i.e., ability of the method to predict key allosteric residues and requirement of sequence depth. For example: the last two paragraphs 565-580 are repetitive as they both highlight the need for depth in sequence alignments which is a known issue for MSA dependent methods, as again discussed by Hopf et al. Nature Biotechnology volume 35, pages 128-135 (2017). Further, the theoretical formulation to extract the coupling score is identical to Hopf et al. and a compare/contrast of the two approaches would be highly beneficial to the audience. Even Hopf et al. highlight the significance of the pairwise term, as done in this article.

---

## [Author Response]

Essential revisions:There seems to be a consensus between the reviewers that:1) There is a lack of comparison of the proposed method against existing ones that do pretty much the same. This is the main weakness of the manuscript.

We agree that comparison of our method with existing ones should be given whenever possible. However, most existing methods based on sequence statistical analysis were developed for predicting allosteric communication pathways. To the best of our knowledge, no systematic computational methods for identifying key allo-residues, residues that contribute most to allosteric signaling in allosteric sites are available now. As the main purpose of our current study is to predict key allo-residues in allosteric sites, direct comparisons with other methods are not possible. As suggested, we have performed indirect comparisons with the Statistical Coupling Analysis (SCA) method, which have been included in the revised manuscript (The last paragraph of KeyAlloSite can also identify key allosteric functional residues of enzymes).

SCA predicts several groups of coevolved residues (Sectors) that form physically continuous networks, often able to connect major functional site and allosteric sites, that is, allosteric pathways. We analyzed all the proteins in our data set using SCA. We first compared the performance of SCA and KeyAlloSite in predicting known key allo-residues in allosteric sites (Supplementary File 9). For the known key allo-residue L359 in the BCR-ABL1 protein, it was not present in the sectors predicted by SCA, despite that the sectors contain 68 residues, while it could be correctly predicted by KeyAlloSite. For the known key allo-residues Y149 and Q152 in the Tar receptor, KeyAlloSite could correctly predict both of them. However, although the sectors predicted by SCA contained Y149 and Q152, it also included two residues (R69 and R73) that have been experimentally verified to contribute only to ligand binding and not to allosteric signaling (*PNAS* 2013, 110, 16814). For PDZ3, the sectors predicted by SCA contained the key allo-residues A347 and L353, which were also successfully predicted by KeyAlloSite. We further compared the performance of SCA and KeyAlloSite in predicting the key allosteric functional residues of enzymes. For the CALB, the sectors predicted by SCA missed one of the key allo-residues A225, which has been experimentally shown to have a great impact on enzyme activity, and the predicted known key allo-residues account for 32.8% of all the residues in the sectors, while KeyAlloSite could predict the key allo-residue A225, and the predicted known key allo-residues account for 38.5% of all the predicted key allo-residues. For the CMs, the sectors predicted by SCA contained only one key allo-residue D83, while KeyAlloSite could predict key allo-residues R44 and L40. For the KeyAlloSite correctly predicted functional phosphorylation sites T288 and T287 in the AurA, SCA missed both of them. Thus, KeyAlloSite performs better than SCA in predicting key allo-residues.

2) The authors should also provide a better metric of the true positives of their method – and of course report the false positives they have for the cases they've tested. AlloSite seems to predict several residues as key, some of which happen to be oncogenic (Table 1). It's not clear how many of the top 10, let's say, residues that it predicts have an allosteric effect.

We have added the confusion matrices of our method in different scenarios (Supplementary file 8) in the revised manuscript. It should be noted that the known real positive data is very limited. At the same time, there is little known real negative data, and we regarded the data of unknown functions as negative data. It can be seen that KeyAlloSite has high recall (0.92 and 1.00) in the prediction of allosteric sites and the prediction of key allo-residues in allosteric pockets. The recall of KeyAlloSite in the prediction of functional post-translational modification (PTM) sites, pathogenic mutations and enzymes are not very high, mainly because there are limited known data on functional PTM sites and pathogenic mutations, and the functions of many key allo-residues predicted by KeyAlloSite are unknown. It remains to be further verified whether they have the function of allosteric regulation. Moreover, since we are now taking the predicted key allo-residues of unknown function as negative samples, our values for recall here are lower bounds of recall. The prediction list of KeyAlloSite provides a good starting point for further experimental studies and we believe that the prediction accuracy would be increased after the functions of predicted residues were experimentally identified.

3) In addition to the comparison, the authors need to establish that the signal coming from comparison of allosteric vs. orthosteric is different from comparing any two random patches with same number of residue pairs.

As suggested, we have added the difference between the evolutionary coupling between orthosteric and allosteric pockets and the evolutionary coupling between two random patches in the protein in the revised manuscript (The last paragraph of The evolutionary coupling between orthosteric and allosteric sites is stronger). For each protein in the data set, two residues that are not part of the orthosteric and allosteric sites were randomly selected from the surface residues. Among them, one was taken as the first center and the residues around it with the same number as the residues in orthosteric pocket were selected as patch1; and the other residue was taken as the second center and the residues around it with the same number as the residues in allosteric pocket were selected as patch2. Then we calculated the evolutionary coupling strength between the patch1 and patch2. The process was repeated four times, and then the mean and standard deviation of the evolutionary coupling strength were calculated. We then compared the evolutionary coupling strength between patch1 and patch2 with that between orthosteric and allosteric sites. The results showed that the evolutionary coupling strength between orthosteric and allosteric sites was significantly higher than that between two random patches (Figure 2—figure supplement 3). In other words, there is intrinsic evolutionary coupling between orthosteric and allosteric sites, which is different from the evolutionary coupling between any two random patches.

4) A complete picture of the predictive power with both strengths and weakness of the tool should be presented.

We have added discussions on the strengths and weaknesses of our approach in the revised manuscript (The second paragraph of *Discussion*). The main strength of our method is that it is the first systematic and efficient computational method to predict key allosteric residues in allosteric sites that are primarily responsible for allosteric signaling. The weakness of our method is that as KeyAlloSite attempts to capture coevolutionary coupling between residues from multiple sequence alignment (MSA), it requires that MSA should contain sufficient homologous and diversified sequences. For the MSAs with only a few homologous sequences, KeyAlloSite usually cannot give accurate predictions. How to reduce the number of homologous sequences required remains further research.

Reviewer #1 (Public Review):Allostery refers to processes whereby a change at one site of a biological macromolecule affects the structure and dynamics at another distinct functional site, enabling the regulation of the corresponding function. Xie et al. developed an in-silico method to predict residues involved in allosteric regulation using a coevolution-based method. A fast and accurate method of identifying key residues responsible for allosteric signalling is important for drug design purposes and protein engineering.

We thank the reviewer for the positive remarks and encouragement.

Strengths:1) The authors applied their method to multiple targets from different protein families to test their method.2) The method is able to predict in a retrospective analysis certain residues involved in allosteric communication between orthosteric and allosteric binding sites.Weaknesses:1) There are several tools used in statistical genomics to predict allosteric communication pathways. Even though the paper tries to demonstrate the ability to predict residues that are involved in allosteric communication, KeyAllosite is not compared with any other state-of-the-art tool that does the same (1-3), which would highlight the strengths of this method with respect to existing ones.

We thank the reviewer for the comment. We have compared our method with statistical coupling analysis method that has been used to predict allosteric communication pathways. Please refer to our response to the first comment in Essential Revisions. We would like to emphasize that our method was developed mainly for predicting key allosteric residues and not for predicting allosteric communication pathways.

2) The authors mention that the number of effective homologue sequences affects the probability of finding an allosteric site in the top 3 scored sites, however, Cdc4 and AR2 have also a low number of effective homologue sequences (Figure 1.A) yet a high Z-score. No sufficient explanation is given regarding this discrepancy. This also demonstrates that a threshold in the Neff under which KeyAlloSite becomes unreliable should be defined.

We thank the reviewer for the comment. A high number of effective homologous sequences guarantee the high probability of the correct prediction of allosteric site. Furthermore, the prediction power also depends on several other factors, such as when the allosteric site appears in evolution, how many effective homologous sequences contain the allosteric site in a particular protein family, and the size of allosteric pockets. We have built the protein sequence based phylogenetic tree of AR1 homologous proteins and found that AR1 located near the tail of the phylogenetic tree (Figure 2—figure supplement 1 in the revised manuscript). This implies that this allosteric function did not exist in the early evolutionary period and the allosteric function may have appeared in the late stage of evolution. Due to the relatively large number of sequences in the early evolutionary period and relatively few sequences in the late stage of evolution, the allosteric signal was weak. For simplicity, we defined the threshold of the number of effective homologous sequences (Neff) as 100 (Section *The allosteric protein data set* in *Methods and Materials*). When Neff is less than 100, the prediction result may become unreliable, but a low number of Neff does not necessarily correspond to a small Z-score. For example, Cdc4 and AR2 have high Z-scores despite their small Neff.

3) From the low Z-score of CYP3A4, the authors claim that the conservation of residues in the orthosteric site is important for getting an accurate prediction of the coupling strength. It is not clear though if and how is this aspect encoded in the KeyAlloSite. From the description of the method, it seems like the algorithm does not check for the level of conservation of the orthosteric residues. An explanation as to why has it not been incorporated into the algorithm is necessary. The conservation at a given site in an MSA defined as the overall deviance of amino acid frequencies at that site from their mean values, in combination with the statistical coupling of two sites has been shown to be important in the development of allosteric models (4).

We thank the reviewer for the comment. Since orthosteric and allosteric sites are functionally coupled and coevolve, we mainly focus on the evolutionary coupling between residues in orthosteric and allosteric sites. We did not explicitly encode the conservation of orthosteric residues in KeyAlloSite as orthosteric residues of most proteins are relatively conserved and ECM already contains sequence conservation information. When the conservation of residues in the orthosteric site varies largely as in the case of CYP3A4, it will be hard to give reliable prediction. We have discussed this issue in the revised manuscript (The first paragraph of The evolutionary coupling between orthosteric and allosteric sites is stronger).

4) In the case of AuroraA, the authors do not explain why other Ser/Thr/Tyr residues scored higher than T287 and T288, or if their higher scores are an artefact. However, in many cases, post-translational modifications take place by secondary partners and, therefore, the coupling of a post-translational modification site with the orthosteric site cannot be used to predict such sites. Post-translational modification sites are expected to have a strong coupling with residues of the upstream/downstream effector that is responsible for the modification, rather than with residues of the orthosteric site of the protein.

We thank the reviewer for the comment. For the three residues T235, S245 and S249 with higher scores than T287 and T288, currently no experimental data are available to verify the effect of their phosphorylation on protein function. Further studies are needed to investigate whether these three residues have allosteric regulatory function, and we think these three residues are very worthy of further study.

It is true in many cases, post-translational modifications (PTMs) take place by secondary partners, and the coupling between a PTM site and residues of upstream/downstream effector determines whether this PTM reaction will happen. However, whether this PTM would influence protein function is encoded in its coupling with the protein orthosteric site as functionally coupled residues usually coevolve. A recent study has also shown that there are strong coevolutionary couplings between post-translational modification sites and orthosteric sites (Zhu, *et al.*, *J Chem Inf Model*. 2022; 62:3331).

5) The authors defined allo-residues as the residues whose Z-score is >0.8, but there is no strong argument regarding the choice of this threshold. In the case of the allosteric pockets, for example, the authors use a threshold of 0 to identify pockets with strong coupling strength.

We thank the reviewer for the comment. We chose this threshold to ensure that most of the known key allo-residues can be correctly predicted by KeyAlloSite, and at the same time, the number of predicted key allo-residues should be as small as possible. We calculated the number of known key allo-residues that could be predicted by KeyAlloSite (Supplementary file 5) and the ratio of predicted key allo-residues in all residues of allosteric pockets of all proteins in the data set (Figure 3—figure supplement 1) for thresholds of 0.5, 0.6, 0.7, 0.8, 0.9, 1.0. It can be seen from the Supplementary file 5 that when taking 0.5, 0.6, 0.7 and 0.8 as thresholds, KeyAlloSite could correctly predict the known key allo-residues in BCR-ABL1, Tar and PDZ3. But when taking 0.9 and 1.0 as thresholds, several known key allo-residues could not be correctly predicted. Moreover, when the threshold was 0.8, the ratio of predicted key allo-residues among all residues of allosteric pockets is less than that of 0.5-0.7 (Figure 3—figure supplement 1). Therefore, we finally chose the threshold as 0.8. We have added this explanation in the revised manuscript (The first paragraph of Coevolution analysis revealed key allo-residues in allosteric pockets).

6) In the case of the Tar, it would be good if the authors reported the results starting from an apo structure as well and see if the method is able to find Y149 and Q152. That way, they could test how sensitive/biased the method is to the chosen conformation. If no apo structure is available, maybe another protein where both an apo and a holo structure exist could be used for this purpose.

We thank the reviewer for the suggestion. We have added the results using the apo structure of Tar in the revised manuscript (The first paragraph of KeyAlloSite correctly identified key allo-residues in other proteins not in the data set). We first used CAVITY to identify all the potential ligand binding pockets on the surface of chain B in the apo structure of Tar (PDB ID: 4Z9J). Of the eight pockets found, cavity_5 is the allosteric pocket, which contains 24 residues. As in the holo structure, the 16 residues (A166-T181) in the C-terminal of the α4 helix were chosen as the orthosteric site. In the prediction results, Y149' and Q152' ranked fourth and fifth among the 24 residues, with corresponding Z-scores of 1.01 and 0.66, respectively. Therefore, KeyAlloSite could also correctly predict the key allo-residue Y149 in the apo structure. For Q152, its Z-score is slightly smaller than the threshold of 0.8, though with a high ranking. This indicates that conformational changes do have subtle influence on the predicted results, probably mainly due to the change of residue composition in the allosteric pocket detected by CAVITY, which will lead to some fluctuations in the predicted key allo-residues. However, when the conformational changes between apo and holo states are not large, the influence on the results is small.

7) The data in Table 1 is not sufficiently convincing. It would be helpful to also report how many of the identified residues by KeyAlloSite are indeed involved in oncogenic mutations or find another metric to quantify the success rate of KeyAlloSite.

We thank the reviewer for the suggestion. Among the 51 KeyAlloSite predicted key allo-residues in 7 human proteins, 11 have been found as oncogenic mutations in the Allo-Mutation database. We have added related information in the revised manuscript (KeyAlloSite identified pathogenetic mutations in human proteins). With the increasing number of known allosteric residues involved in oncogenic mutations, we expect that KeyAlloSite can identify more pathogenetic mutations.

8) The success rate of the method to predict key residues of the function of CALB (38% success rate based on the reported residues in the literature) and CMS (20%), should be interpreted with caution. It may be the case that the rest of the predicted residues have not been tested for their functional role, or that the method has indeed a low success rate in predicting key residues for allosteric communication.

We thank the reviewer for the suggestion. Wu *et al.* identified 63 annotated functional residues in CALB from literature (*FASEB J* 2020, 34, 1983), of which 8 were in the substrate binding pocket and the remaining 55 annotated functional residues were outside the substrate binding pocket. There are 296 residues outside the substrate binding pocket, and the annotated functional residues account for 18.6% of all residues, so it is difficult to predict these annotated functional residues. Wu *et al.* developed the SCA.SIM method and found that the success rate of SCA.SIM was better than that of SCA. The top 27 ranked residues predicted by SCA.SIM contained 11 annotated functional residues, 6 of which are in the substrate binding pocket and 5 of which are outside the substrate binding pocket. This means that the predicted success rate of the annotated functional residues outside the substrate binding pocket is 18.5%. In comparison, the top 27 ranked residues predicted by KeyAlloSite contained 7 annotated functional residues outside of the substrate binding pocket with a success rate of 25.9%. The top 38 ranked residues predicted by SCA.SIM contained 12 annotated functional residues, 6 of which are in the substrate binding pocket and 6 of which are outside the substrate binding pocket with a prediction success rate of 15.8%. In comparison, the top 38 ranked residues predicted by KeyAlloSite contained 9 annotated functional residues outside of the substrate binding pocket with a success rate of 23.7%. Therefore, the success rates of KeyAlloSite was slightly higher than that of SCA.SIM in predicting annotated functional residues outside of the substrate binding pocket. For the rest of the predicted residues that have not been tested for their functional role, we reason that they are likely to allosterically regulate the activity of the enzyme, and further experiments are needed to test whether they really have allosteric regulatory functions.

In lines 38-41, the authors refer to SARs of allosteric drugs as "flat", which is not clear what they refer to.

“Flat” SARs (also sometimes referred to as “Shallow” SARs) means that SARs are not robust and a modest structural modification may destroy the activity of a potent allosteric modulator. More explanations on this have been added in the revised manuscript.

It is not clear if the so-called, "allo-residues", that the authors define in lines 49-52 refer to residues that affect the binding or the signalling.

“Allo-residues” refer to residues that mainly affect signaling. More explanations on this aspect have been added in the revised manuscript.

It would be helpful for the reader if the authors included a section where they describe the method itself and the key steps of developing their model before presenting the results. That way, the reader could follow the results easier.

We have added a section describing the method and the key steps of developing the model before presenting the results in the revised manuscript (The last paragraph of the *Introduction*).

The authors classified L359 of BCR-ABL1 as an allo-residue and justified the importance of this residue based on the fact that a weaker ligand does interact with this residue. For transparency, it would be good for the authors to report the allosteric scores of all 44 residues in the allosteric site to show that this residue was not cherry-picked and that the method can pick up all the residues that are important for the binding.

We have added the Supplementary file 6, which reported the allosteric scores of all 44 residues in the allosteric site.

The manuscript has several typos and language mistakes (e.g. "screening" instead of "screen" in the Abstract, missing articles, etc.) that should be corrected prior to a resubmission.

We have carefully checked the language and made necessary corrections in the revised manuscript.

Reviewer #2 (Public Review):The authors designed a statistical approach to analyse coevolution scores from a protein and predict allosteric residues. This approach relies on comparison of residues from the two sites (allosteric vs. orthosteric) and omits the rest of the protein.While the approach is logical, the predictive power has not been clearly established. To demonstrate the effectiveness of the approach a confusion matrix should be provided as it will show the predictive power of the method in the different scenarios presented in the manuscript (PTM's, enzymes, pathogenic mutations, etc.).An effective method, along these lines, will have significant impact on protein and drug design.

We thank the reviewer for the comments and encouragement. To demonstrate the predictive power of our method, we have given the confusion matrices in different scenarios (Supplementary file 8) in the revised manuscript. In different scenarios, the known real positive data is very limited. At the same time, there is little real negative data, and we regarded data with unknown functions as negative data. It can be seen that KeyAlloSite has high recall in the prediction of allosteric sites and the prediction of key allo-residues in allosteric pockets. The recall of KeyAlloSite in the prediction of functional post-translational modification (PTM) sites, pathogenic mutations and enzymes are not very high, mainly because there are limited known data on functional PTM sites and pathogenic mutations, and the functions of many key allo-residues predicted by KeyAlloSite are unknown. It remains to be further verified whether they have the function of allosteric regulation. Moreover, since we are now taking the predicted key allo-residues of unknown function as negative samples, our values for recall here are lower bounds of recall. The prediction recall of KeyAlloSite will be higher if the residue of the unknown function we predicted is later determined to have an allosteric regulation function.

It would be helpful if statistics on the distribution of significant scores from E row comparison after the t-test are provided. Further, you have so far compared allosteric with orthosteric, it would be interesting to compare the same distribution/statistics with allosteric vs. non-orthosteric residues. It may serve as a very good benchmark. The idea would be to see if the events of significance from allosteric vs. orthosteric are different/unique if we pick and compare any two random patches in the protein.

As suggested, we have added some examples of distributions of the statistics corresponding to significant scores obtained from the t-test (Figure 3—figure supplement 2).

We also have added the difference between the evolutionary coupling between orthosteric and allosteric pockets and the evolutionary coupling between two random patches in the protein in the revised manuscript (The last paragraph of The evolutionary coupling between orthosteric and allosteric sites is stronger). For each protein in the data set, two residues that are not part of the orthosteric and allosteric sites were randomly selected from the surface residues. Among them, one was taken as the first center and the residues around it with the same number as the residues in orthosteric pocket were selected as patch1; and the other residue was taken as the second center and the residues around it with the same number as the residues in allosteric pocket were selected as patch2. Then we calculated the evolutionary coupling strength between the patch1 and patch2. The process was repeated four times, and then the mean and standard deviation of the evolutionary coupling strength were calculated. We then compared the evolutionary coupling strength between patch1 and patch2 with that between orthosteric and allosteric sites. The results showed that the evolutionary coupling strength between orthosteric and allosteric sites was significantly higher than that between two random patches (Figure 2—figure supplement 3). In other words, there is intrinsic evolutionary coupling between orthosteric and allosteric sites, which is different from the evolutionary coupling between any two random patches.

A boxplot of Zscores for All, top200, top300, top400 pairs from Figure 1—figure supplement 2 would be very informative.

We have added the boxplot of Z-scores for All, top200, top300, top400 pairs from Figure 1—figure supplement 2 as suggested (Figure 2—figure supplement 2C).

As you have been using top zscore residues (>0.5 or >0.9 for enzymes), direct or indirect evidence for the statement in lines 306-307 is not very apparent from the provided data. This has to be flushed out further.

Thank you for the suggestion. We feel sorry for not explaining clearly how weak terms in Jij were used in our prediction. Since orthosteric and allosteric pockets are two different pockets, the residues in the two pockets generally do not contact. Therefore, when we calculate the coevolutionary coupling between residues in the two pockets, most of the evolutionary coupling values between residues in the two pockets are relatively small, that is, they correspond to the weak terms in Jij, as can be seen in Figure 1D. Then we got the number of significant differences of each residue by comparing the differences of these weak evolutionary coupling values of each residue in the allosteric pocket, and normalized the number of significant differences to Z-scores. A larger Z-score indicates that the evolutionary coupling values of this residue are significantly different from the evolutionary coupling values of more residues. In other words, Jij and Z-score are two different metrics. We have added this detailed explanation of the use of the weak terms in Jij in the revised manuscript (The first paragraph of Coevolution analysis revealed key allo-residues in allosteric pockets).

Please annotate residues for which experimental data is available to compare with keyallosite predictions for example in Supplementary files 4 and 5. In the main text (line 284) you only mention one residue out of 52 to demonstrate the effectiveness of the prediction. Again a confusion matrix would help here.

Thank you for the suggestion. We have bolded the predicted key allo-residues that were annotated as functional residues by the experimental data in the previous literature in the revised Supplementary files 4 and 7 as suggested. In the main text (line 284), V225M is a representative of the functional residues already annotated with experimental data in CALB, and other functional residues annotated with experimental data are included in revised Supplementary file 7. We have added another example, V37I, and the prediction accuracy calculated based on confusion matrix was 74.0%. Although the prediction accuracy here is not very high, it is the lower bound of the prediction accuracy of KeyAlloSite. The prediction accuracy of KeyAlloSite will be higher if the residue of the unknown function we predicted is later determined to have an allosteric regulation function.

Can you please clarify if in Figure 2, number of residues refers to the number of residues from allosteric+orthosteric site?

Thank you for pointing this out. The number of residues refers to the number of residues from allosteric site. We have added this detailed description in the revised Figure which is now Figure 3.

Figure 6 may be ideal as Figure 1 to inform the readers about the protocol of this work.

Thank you for the suggestion. We have adjusted Figure 6 as Figure 1 in the revised manuscript as suggested.

[Editors' note: further revisions were suggested prior to acceptance, as described below.]

The manuscript has been improved but there are some remaining issues that need to be addressed, as outlined below:

We thank the Senior Editor, the Reviewing Editor and the reviewers for the comments and suggestions which are helpful to improve our work.

1. Can you please provide a detailed description and expand the discussion on the predicted amino acids and why the predicted residues are not the top ranking ones?

Thank you for the suggestion. We have added the detailed description and discussion on the predicted other top-ranking key allo-residues with unidentified function (The second paragraph of *Discussion*), the comparison to the approach of Hopf *et al.* (The third paragraph of *Discussion*), the possible effect of protein conformational changes on the prediction results (The fifth paragraph of *Discussion*) and possible applications of the predicted key allo-residues (The last paragraph of *Discussion*) in the revised manuscript.

2. The theoretical formulation to extract the coupling score and a compare/contrast to Hopf et al. approach would be highly beneficial to the readers.

We have added the theoretical formulation for extracting the coupling scores (The first paragraph of The Evolutionary Coupling Model (ECM)) and the comparison to the approach of Hopf et al. (The third paragraph of Discussion) in the revised manuscript. The theoretical formulation for extracting the coupling scores between residues in the initial step of our method is the same as that in the method of Hopf et al., but the problem studied by us is different from that of Hopf et al., and the usage of the coupling scores between residues in the later steps is different. Hopf et al. used the coevolutionary coupling scores between residues to predict the effects of mutations by calculating the difference in statistical energy between mutant and wild-type sequences. In contrast, we used the coevolutionary coupling scores between residues to predict allosteric sites and predict the key allo-residues in allosteric pockets that are mainly responsible for allosteric signaling by pairwise comparing the difference of the coevolutionary coupling scores of residues in allosteric pockets. Although Hopf et al. highlighted the significance of the pairwise term for the prediction of mutation effects, we highlighted the importance of the weak pairwise term for the study of allosteric function.

Reviewer #1 (Recommendations for the authors):The authors did extensive work trying to address the revision comments, and their effort is much appreciated.Nevertheless, in the absence of further experimental validation of the importance of the predicted so-called key "allo-residues" in mediating the signal between the orthosteric and allosteric binding site, it is still hard to assess the predictive power of the presented method. There are still residues for example that score higher than known functional residues (see for example residues T235, S245 and S249 in the case of Aurora A) whose implication in signal transduction is not confirmed, or residues whose importance depends on the conformation chosen for analysis despite the coevolutionary conservation metric used to score the residues (Q152 of Tar whose score is way below the 0.8 threshold when an apo structure is considered, which shouldn't be the case given that AlloSite uses only MSA and coevolutionary information for the scoring – it seems like other residues are predicted to be way more important based on the allosteric pocket definition given by CAVITY to decrease the score of Q152 to <0.8 after normalisation).I fully understand that such experimental validation goes beyond the capacities of a computational group, however, as a reader, I might be hesitant to try this method.

We thank the reviewer for the comments. We agree that experimental validation of the newly predicted key "allo-residues" in mediating allosteric signaling will further verify the predictive power of our method. We are planning to carry out experimental validation studies in the future. Currently we only have a small number of proteins with limited experimental information about key allo-residues. Among these proteins, our predicted key allo-residues rank high among all the alloresidues. At the same time, there are residues with higher scores than these known functional residues, which may play important roles in allosteric signaling. Of course, further experimental validation will be necessary to study the function of these top-ranking residues. With more experimental data at hand, the method can be further validated and improved in the future.

The scores of the key allo-residues predicted by KeyAlloSite depend not only on the coevolutionary information, but also on the residues that constitute the allosteric pockets (Figure 1, see *Materials and methods* for details). For Tar, the evolutionary couplings between residues are the same for the apo and holo conformations, but the allosteric pockets found by CAVITY in the two conformations contain a number of different residues. Because the prediction of key allo-residues by KeyAlloSite requires pairwise comparison of residues in allosteric pockets, the predicted key allo-residues in the two conformations were slightly different. For the apo Tar, on the one hand, although the score of Q152 is 0.66, which is less than the threshold of 0.8, Q152 ranked high among all residues in the allosteric pocket with a ranking of 5/24. When we lower the threshold slightly, we will be able to correctly predict Q152. On the other hand, Bi *et al.* showed that although Y149 and Q152 are both key allo-residues, Y149 seems to be more important as allosteric signaling can be conducted when the allosteric molecule only interacts with it and not Q152 (*PNAS* 2013 110:16814-16819. DOI: https://doi.org/10.1073/pnas.1306811110). Therefore, although Q152 was not predicted in the apo Tar, allosteric molecules can still be optimized based on the Y149 predicted by KeyAlloSite.

Reviewer #2 (Recommendations for the authors):The authors have carefully addressed all the comments from the previous review.Few additional comments for authors to consider:The Discussion section can be enriched further. At present it only discusses two points, i.e., ability of the method to predict key allosteric residues and requirement of sequence depth. For example: the last two paragraphs 565-580 are repetitive as they both highlight the need for depth in sequence alignments which is a known issue for MSA dependent methods, as again discussed by Hopf et al. Nature Biotechnology volume 35, pages 128-135 (2017). Further, the theoretical formulation to extract the coupling score is identical to Hopf et al. and a compare/contrast of the two approaches would be highly beneficial to the audience. Even Hopf et al. highlight the significance of the pairwise term, as done in this article.

We thank the reviewer for the suggestion. We have added the discussion on the predicted other topranking key allo-residues with unidentified function (The second paragraph of *Discussion*), the comparison to the approach of Hopf *et al.* (The third paragraph of *Discussion*), the possible effect of protein conformational changes on the prediction results (The fifth paragraph of *Discussion*) and the utility of the predicted key allo-residues (The last paragraph of *Discussion*) in the revised manuscript.

For the last two paragraphs, although they both mentioned the need for depth in sequence alignments, the two paragraphs emphasized different points. The first paragraph emphasized the need for sufficient homologous sequences, while the second paragraph emphasized that although our method needs to use the three-dimensional structure of a protein, when a protein does not have a known three-dimensional structure but has a certain number of homologous sequences, our method can also be applied.

The theoretical formulation for extracting the coupling scores between residues in the initial step of our method is the same as that in the method of Hopf *et al.*, but the problem studied by us is different from that of Hopf *et al.*, and the usage of the coupling scores between residues in the later steps is different. Hopf *et al.* used the coevolutionary coupling scores between residues to predict the effects of mutations by calculating the difference in statistical energy between mutant and wild-type sequences. In contrast, we used the coevolutionary coupling scores between residues to predict allosteric sites and predict the key allo-residues in allosteric pockets that are mainly responsible for allosteric signaling by pairwise comparing the difference of the evolutionary coupling scores of residues in allosteric pockets. Although Hopf *et al.* highlighted the significance of the pairwise term for the prediction of mutation effects, we highlighted the importance of the weak pairwise term for the study of allosteric function.